# Differential regulation of β-catenin-mediated transcription via N- and C-terminal co-factors governs identity of murine intestinal epithelial stem cells

Costanza Borrelli[1,2,5], Tomas Valenta [1,3,5✉], Kristina Handler[2], Karelia Vélez[2], Alessandra Gurtner[4], Giulia Moro [1], Atefeh Lafzi[2], Laura de Vargas Roditi[2], George Hausmann [1], Isabelle C. Arnold[4], Andreas E. Moor [2] & Konrad Basler [1✉]

The homeostasis of the gut epithelium relies upon continuous renewal and proliferation of crypt-resident intestinal epithelial stem cells (IESCs). Wnt/β-catenin signaling is required for IESC maintenance, however, it remains unclear how this pathway selectively governs the identity and proliferative decisions of IESCs. Here, we took advantage of knock-in mice harboring transgenic β-catenin alleles with mutations that specifically impair the recruitment of N- or C-terminal transcriptional co-factors. We show that C-terminally-recruited transcriptional co-factors of β-catenin act as all-or-nothing regulators of Wnt-target gene expression. Blocking their interactions with β-catenin rapidly induces loss of IESCs and intestinal homeostasis. Conversely, N-terminally recruited co-factors fine-tune β-catenin's transcriptional output to ensure proper self-renewal and proliferative behaviour of IESCs. Impairment of N-terminal interactions triggers transient hyperproliferation of IESCs, eventually resulting in exhaustion of the self-renewing stem cell pool. IESC mis-differentiation, accompanied by unfolded protein response stress and immune infiltration, results in a process resembling aberrant "villisation" of intestinal crypts. Our data suggest that IESC-specific Wnt/β-catenin output requires selective modulation of gene expression by transcriptional co-factors.

[1] Department of Molecular Life Sciences, University of Zurich, Zurich, Switzerland. [2] Department of Biosystems Science and Engineering, ETH Zurich, Basel, Switzerland. [3] Institute of Molecular Genetics of the ASCR, v. v. i., Prague 4, Czech Republic. [4] Institute of Experimental Immunology, University of Zurich, Zurich, Switzerland. [5] These authors contributed equally: Costanza Borrelli, Tomas Valenta. ✉email: tomas.valenta@mls.uzh.ch; konrad.basler@mls.uzh.ch

Intestinal epithelial stem cells (IESCs) fuel the perpetual replenishment of the epithelial monolayer that lines the gut. They divide on average once a day to give rise to fast cycling progenitors (transit amplifiers, TA), which undergo several rounds of cell division before differentiating into cells of the absorptive (enterocytes) and the secretory lineage (Paneth, goblet, tuft, and enteroendocrine cells)[1]. Wnt/β-catenin signalling is required for the maintenance of the IESC pool[2]. However, to date, the gene regulatory networks governing the behavior of IESCs remain elusive. The activity level of the Wnt pathway alone cannot explain the cell-type-specific proliferative rate and self-renewal ability of IESCs, TAs, and Paneth cells[3,4]. Hence, differential regulation of Wnt-target gene expression may determine the identity of the self-renewing pool of IESCs.

A common extracellular signal may be translated into distinct cellular responses, and fate decisions, via the expression of specific cell-autonomous effectors. In the context of Wnt signalling, transcriptional co-factors that interact with β-catenin have been repeatedly implicated in tissue- and cell type-specific modulation of Wnt responses[5]. C-terminal β-catenin interactors, such as CREB-binding protein (CBP) and p300, as well as members of the Mediator and of the RNA polymerase II-associated factor 1 (PAF1) complexes, are promiscuous co-activators that interact, besides with β-catenin, with several other transcription factors[6,7]. Conversely, B-cell lymphoma 9 (BCL9) and the paralog BCL9-like (BCL9L) specifically bind an N-terminal moiety of β-catenin, and act as tissue-specific transcriptional effectors[8–12]. While dispensable for normal intestinal homeostasis, BCL9/9L are required for intestinal regeneration upon insults[13–16], suggesting that they might be necessary for the reconstitution of the stem cell pool. Moreover, compelling evidence has been provided that BCL9/9L play a crucial role in maintaining tumor stemness in several murine models of colorectal cancer[13–16].

It remains untested whether differential regulation of Wnt-signaling outputs is connected to IESC-specific fate determination and self-renewal ability. In this study, we dissect the individual contributions of the C- and N-terminal transcriptional branches of β-catenin to the maintenance of intestinal homeostasis. Using mutant knock-in β-catenin alleles that specifically impair the recruitment of N- or C-terminal β-catenin's co-factors[8], we show how differential regulation of Wnt/β-catenin signalling does govern IESC identity and fate. While C-terminal co-factors act as an all-or-nothing switch for Wnt-target gene expression, N-terminal co-factors are responsible for selective transcriptional modulation ensuring proper proliferation and self-renewal of IESCs.

## Results

### Attenuation of N- vs. C-terminal β-catenin transcriptional outputs in the intestinal epithelium has distinct phenotypic impacts.

We previously generated transgenic β-catenin (Ctnnb1) alleles harboring mutations that prevent interactions with N- or C-terminal transcriptional co-factors (NTFs and CTFs, respectively)[8]. The D164A mutation abrogates interaction with NTFs, while the ΔC truncation abrogates the interaction with the CTFs (Fig. 1a). To overcome the embryonic lethality of these alleles, we used compound heterozygous mice carrying one mutant and one conditional β-catenin allele (Supplementary Fig. 1a). Similarly, to constitutively hemizygous Ctnnb1$^{KO/wt}$ mice, Ctnnb1$^{D164A/flox}$ and Ctnnb1$^{ΔC/flox}$ animals show no overt abnormalities, indicating haplosufficiency of β-catenin for the maintenance of intestinal homeostasis. Combination with the villin-CreER$^{T2}$ driver[17] enables the inducible deletion of the conditional β-catenin allele (Ctnnb1$^{flox}$) specifically in the intestinal epithelium, thus leaving the Wnt-transcriptional outputs solely under the control of mutant β-catenin (D164A or ΔC).

While villin-CreER$^{T2}$;Ctnnb1$^{wt/flox}$ (control) mice are viable and indistinguishable from homozygous wild-type (wt) animals, the sole presence of mutant β-catenin is lethal. Villin-CreER$^{T2}$;Ctnnb1$^{ΔC/flox}$ (ΔC) animals exhibit atrophic crypts and reach humane endpoint 4 days after CreER$^{T2}$ induction (4d post-induction (pi)) (Supplementary Fig. 1b). This is in accordance with our previous results in villin-CreER$^{T2}$;Ctnnb1$^{dm/flox}$ animals, which express double mutant (dm) β-catenin harboring both N- and C-terminal mutations[18]. Villin-CreER$^{T2}$;Ctnnb1$^{D164A/flox}$ (D164A) animals only reach humane endpoint at 7d pi, and suffer from severe colitis (Supplementary Fig. 1c). Thus, neither C- nor N-terminally mutated β-catenin is haplosufficient in the intestinal epithelium, as these mutants are not able to substitute for the wt allele. However, the distinct phenotypic impacts of these mutations suggest different roles of the C- and N-terminal branches of β-catenin-mediated transcription, which we set out to investigate.

We confirmed full recombination of the floxed β-catenin allele in the mutant intestinal epithelium 2d pi (Supplementary Fig. 1d). Consequently, and consistent with the rapid turnover of intestinal epithelial cells, we observed depletion of wt β-catenin protein from crypts 2d pi (Supplementary Fig. 1e). Thus, 2d pi, crypts of ΔC, D164A, and dm animals only harbor mutant β-catenin, which is present at cytosolic and nuclear levels comparable to those of wt β-catenin (Supplementary Fig. 1f). Importantly, crypt and villus integrity remained unaltered. Indeed, as previously reported for the β-catenin-dm animals[8,18], the observed phenotype is entirely connected to transcriptional outputs, and not attributable to loss of epithelial adhesiveness, as in the case of complete β-catenin loss. In fact, mutant β-catenin co-localizes with epithelial cell adhesion molecule (Epcam) at the cell membrane (Supplementary Fig. 1g).

We performed RNA sequencing of bulk preparations of small intestinal epithelium isolated 2d pi from control, ΔC, D164A, dm, and KO animals. Principal component analysis indicated surprising differences in impairing N- versus C-terminal interactions on the epithelial transcriptome (Fig. 1b). The differentially expressed genes (DEGs, logFC > |2|, p < 0.05) of ΔC animals broadly overlapped with those of dm and full knock-out (KO) animals, indicating that C-terminally mutated β-catenin is not able to sustain Wnt signalling levels required for proper renewal of the intestinal epithelium (Fig. 1c and Supplementary Fig. 2a). Indeed, attenuating C-terminal β-catenin transcriptional outputs in the intestinal epithelium causes a loss of IESCs, and attrition of proliferation in the transient amplifier compartment, as evidenced by IESC gene and proliferative marker expression (Fig. 1d), gene set enrichment analysis (GSEA, Supplementary Fig. 2b)[19], as well as protein stainings and RNA in situ hybridization (Fig. 1e–g). Downregulation of bona fide Wnt target genes in ΔC, dm, and KO animals 2d pi was confirmed by qRT-PCR (Supplementary Fig. 2c).

Contrary to what was observed in ΔC mice, the transcriptomic changes induced in β-catenin-D164A animals (i.e., N-terminal mutant) only minimally overlapped with those induced by the loss of β-catenin (Fig. 1c and Supplementary Fig. 2a). Of note, the exclusivity of DEGs in D164A-mutants can be partially attributed to a D164A-specific enrichment of genes expressed by infiltrating immune cells (Supplementary Fig. 2d). As opposed to what we observed in ΔC crypts, the expression of Wnt targets, IESC genes and proliferation markers, was significantly increased in D164A crypts 2d pi (Fig. 1e, f, and Supplementary Fig. 2c). Moreover, D164A mutant crypts displayed an increase of the Lgr5 + area, as shown by single-molecule fluorescent in situ hybridization (smFISH), indicating an expansion of the IESC compartment

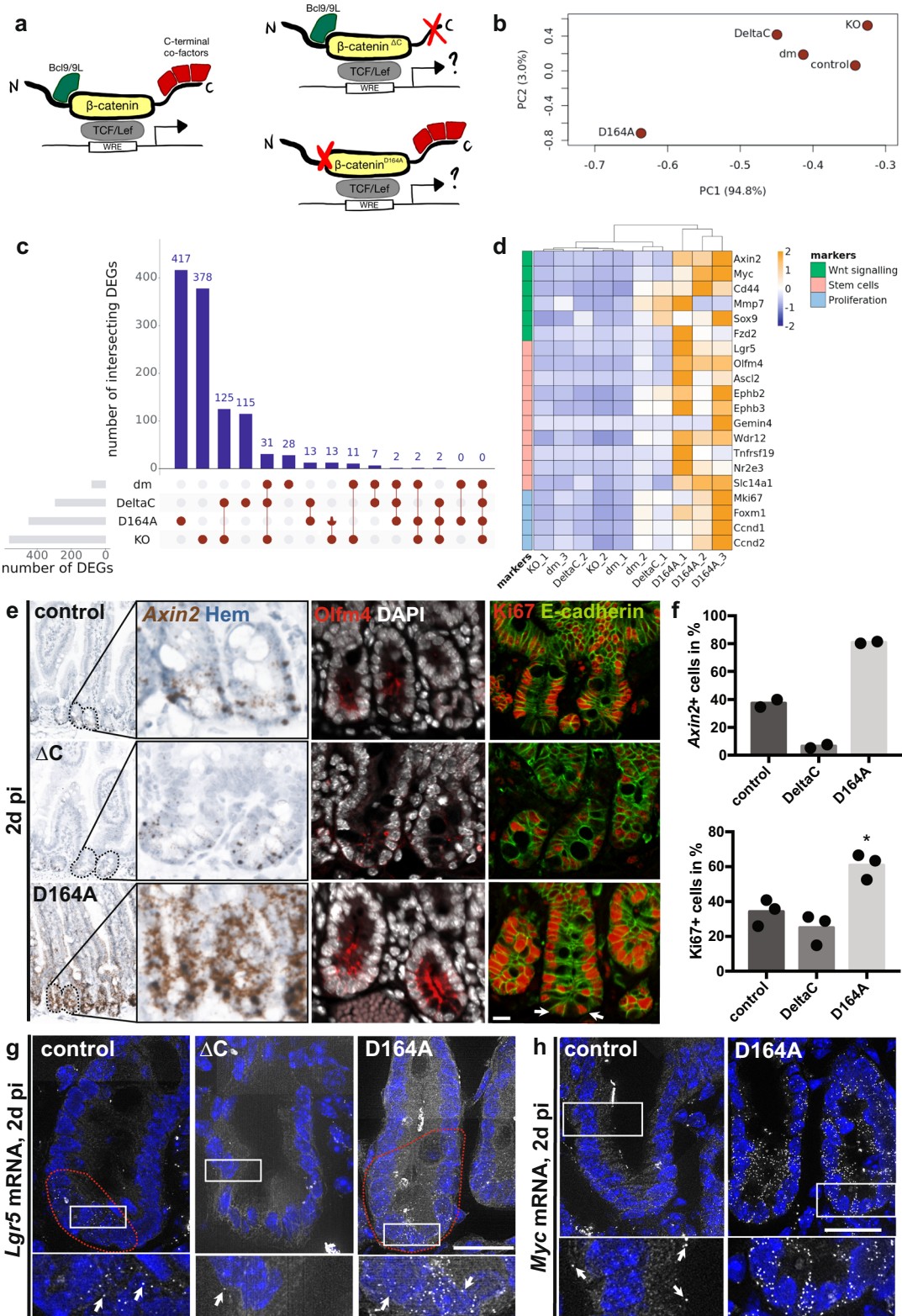

(Fig. 1g and Supplementary Fig. 2e). Furthermore, both the presence of Ki67+ cells at the bottom of D164A crypts (white arrows in Fig. 1e, and evidenced in Supplementary Fig. 2f) and upregulated *Myc* expression at crypt base (Fig. 1h) indicate increased IESC proliferation. These results indicate that preventing β-catenin's interactions to CTFs completely represses the Wnt-outputs, including proliferation and IESC-associated genes, hence compromising stem cell maintenance. On the contrary,

attenuating β-catenin's N-terminal transcriptional outputs increases the proliferation of IESCs and results in an expansion of the stem cell compartment.

Of note, despite the *villin*-driven CreER[T2] being active along the gastrointestinal epithelium, we found neither morphologic nor proliferative changes in the colonic epithelium of β-catenin mutant animals at 2 or 4d pi (Supplementary Fig. 3a, b). This is most likely due to the slower renewal dynamics of the colonic

**Fig. 1 Attenuation of N- vs. C-terminal β-catenin transcriptional outputs has contrasting effects on intestinal homeostasis. a** Scheme of wt and mutant β-catenin proteins with impaired interactions to N- or C-terminal transcriptional co-factors. **b** Principal component analysis (PCA) of the transcriptome of averaged mutant and control small intestinal epithelial RNASeq samples. Timepoint: 2d pi. PC1 and PC2 explain 94.8 and 3% of the variance, respectively. KO: *villin-CreER^{T2};Ctnnb1^{flox/flox}*, n = 2; dm: *villin-CreER^{T2};Ctnnb1^{dm/flox}*, n = 3; control: *villin-CreER^{T2};Ctnnb1^{wt/flox}*, n = 3; D164A: *villin-CreER^{T2};Ctnnb1^{D164A/flox}*, n = 3; ΔC: *villin-CreER^{T2}; Ctnnb1^{ΔC/flox}*, n = 2. **c** Upset plot of intersecting differentially expressed genes (DEGs) (logFC > |2|, p < 0.01, as calculated by edgeR's exact test) in ΔC, dm, KO and D164A, with respect to control. **d** Heatmap of normalized FPKM of Wnt and IESC marker genes across hierarchically clustered mutants. Rows are scaled. **e** *Axin2* (Wnt-target) mRNA in situ hybridization, Olfm4 (IESC marker) and Ki67 (proliferating cells) immunofluorescence of control, D164A and ΔC duodenal crypts. Hematoxylin, DAPI (nuclei) or E-cadherin (cell membrane) as counterstain. White arrows indicate Ki67+ cells at the crypt base. Timepoint: 2d pi. Scale bar, 20 μM. **f** Quantification of Axin2+ cells (n = 2) and Ki67+ cells (n = 3) per crypt. *p = 0.019, as calculated by unpaired, two-tailed Student's T-test. Barplot indicates the mean number of positive cells. **g, h** smFISH of *Lgr5* and *Myc* mRNA. Insets show higher magnification of crypt base. Arrows indicate single mRNA molecules visible as dots. Red dashed line indicates Lgr5+ stem cell compartment. Timepoint: 2d pi. Scale bar, 20 μM. Representative images of three biological replicates.

epithelium[20]. Moreover, high E-cadherin levels in the colon act as a sink for β-catenin, delaying the effects of introduced mutations[21]. We therefore focused our investigation on the small intestinal epithelium.

**Intestinal crypts lacking the output via N-terminal recruited co-factors exhibit stem cell loss and secretory hyperplasia 4d pi.** We had not observed any overt phenotype of goblet cells or Paneth cells in ΔC or D164A animals 2d pi (Supplementary Fig. 3c, d). Unexpectedly, 4d pi, crypt morphology in D164A was profoundly altered by the appearance of large granular cells that lacked expression of Wnt-target genes (*Axin2*), as well as proliferative (Ki67) and IESC (Olfm4) markers (Fig. 2a). These cells express the secretory lineage marker Sox9 (Fig. 2a), and are double positive for lysozyme (Lyz) and mucins (stained by alcian blue, AB) (Fig. 2b). Moreover, we observed Lyz⁺ cells mislocalized to the villus of D164A animals (Fig. 2b). These cells are reminiscent of intermediate cells, which share both Paneth (lysozyme) and goblet cell traits (mucins), are the result of stem cell mis-differentiation, and have been observed upon experimental perturbation of Notch, Wnt, or EGF signalling in the intestinal epithelium[22–26]. However, the molecular mechanisms underlying their appearance in the tissue is largely unknown. Interestingly, a shift towards secretory lineage differentiation was observed upon deletion of N-terminal co-factors BCL9/9L in APC^{min} tumors[16], suggesting that NTFs might repress specification of the secretory lineage.

Taken together, the results presented so far indicate that, similar to the complete deletion of β-catenin, blocking its CTF recruitment rapidly caused a lethal loss of stem cells (Fig. 2C). On the other hand, impairing interaction with BCL9/9L triggers transient hyperproliferation of IESCs, followed by their aberrant differentiation. The profoundly distinct cellular dynamics leading to crypt collapse upon impairment of N- vs. C-terminal β-catenin outputs, thus suggest independent roles of these transcriptional branches in maintaining IESC homeostasis.

**Hyperproliferative IESCs expressing only *Ctnnb1^{D164A}* have an increased ability to form organoids.** Recent work indicated that intestinal organoids could not be established from BCL9/9L-deficient crypts isolated 4d pi[15]. However, we hypothesized that expansion of the proliferative IESC compartment in D164A crypts observed 2d pi would translate into increased organoid formation rate. To test this, we devised an organoid formation assay (Fig. 3a): we isolated intestinal crypts from *villin-CreER^{T2}; Ctnnb1^{wt/flox}* (control) and *villin-CreER^{T2};Ctnnb1^{D164A/flox}* (D164A) animals 0, 2, and 4d post CreER^{T2} induction, seeded equal amounts in Matrigel and quantified the number of organoids formed 7 days later. Strikingly, D164A crypts isolated 2d pi exhibited a 1.6-fold increase in organoid formation rate compared

to non-induced (*Ctnnb1^{D164A/flox}*) crypts, and a 10-fold increase compared to the respective control (Fig. 3b). Morphologically, organoids derived from hyperproliferative crypts showed increased cell proliferation and budding (Fig. 3c). Consistent with the transient nature of the hyperproliferation observed in vivo, organoid lines obtained from D164A crypts 2d pi cannot be maintained in culture, and died upon first passaging (post splitting panels, Fig. 3c). Crypts isolated from D164A animals 4d pi failed to establish organoids in vitro, in agreement with the loss of stem cell and proliferation markers observed in the tissue (Fig. 3b). In sum, these results in vitro recapitulate and functionally underscore our observations in vivo.

We wondered why control crypts isolated 2d and 4d pi showed a drastic reduction in organoid formation rate compared to unrecombined crypts (Fig. 3b). We found that, in contrast to the in vivo situation, β-catenin is not haplosufficient in vitro: crypts isolated from constitutively hemizygous mice (*Ctnnb1^{KO/wt}*), which show no overt phenotype, fail to establish intestinal organoids (Supplementary Fig. 3a). Moreover, crypts isolated from *Ctnnb1^{ΔC/flox}* and *Ctnnb1^{dm/flox}* animals also fail to grow in vitro (Supplementary Fig. 4a), indicating that two copies of transcriptionally active β-catenin alleles are necessary to maintain Wnt-target gene expression required by intestinal organoids. We established organoid cultures from non-injected *villin-CreER^{T2}; Ctnnb1^{wt/flox}* (control) and *villin-CreER^{T2};Ctnnb1^{D164A/flox}* (D164A) animals, and induced recombination in vitro. In contrast to organoids lacking the *villin-CreER^{T2}* driver, both control and D164A organoids died within 7 days after 4OHT addition, and quickly lost expression of markers indicative of Wnt signalling, stemness or proliferation (Supplementary Fig. 4b–d). This behavior could not be rescued by the addition of Wnt3a-conditioned medium to the culture prior to 4OHT addition (Supplementary Fig. 4e). The differential Wnt/β-catenin-signalling requirements of intestinal crypts in vitro might be due to the fact that organoids lack a mesenchymal support niche, and that they closely resemble a regenerative, rather than a homeostatic state of intestinal crypts.

**Myc-E2F-driven proliferation leads to exhaustion of the stem cell pool in N-terminal mutant (D164A) animals.** To dissect the gene regulatory networks underlying the transient hyperproliferation of IESCs triggered by impaired β-catenin-NTF interactions, we performed droplet-based single-cell RNA sequencing (scRNAseq) of small intestinal crypts isolated from control and D164A animals at 0, 2, and 4d pi. We focused our analysis on the gene expression changes occurring over time in stem cells and early progenitors (SCEP cells), which we annotated manually based on marker gene expression (Supplementary Fig. 5a, b). Differential gene expression analysis confirmed our previous results, indicating that IESCs and proliferation markers (e.g., *Olfm4* and *Mki67*) are upregulated in D164A SCEP cells 2d pi

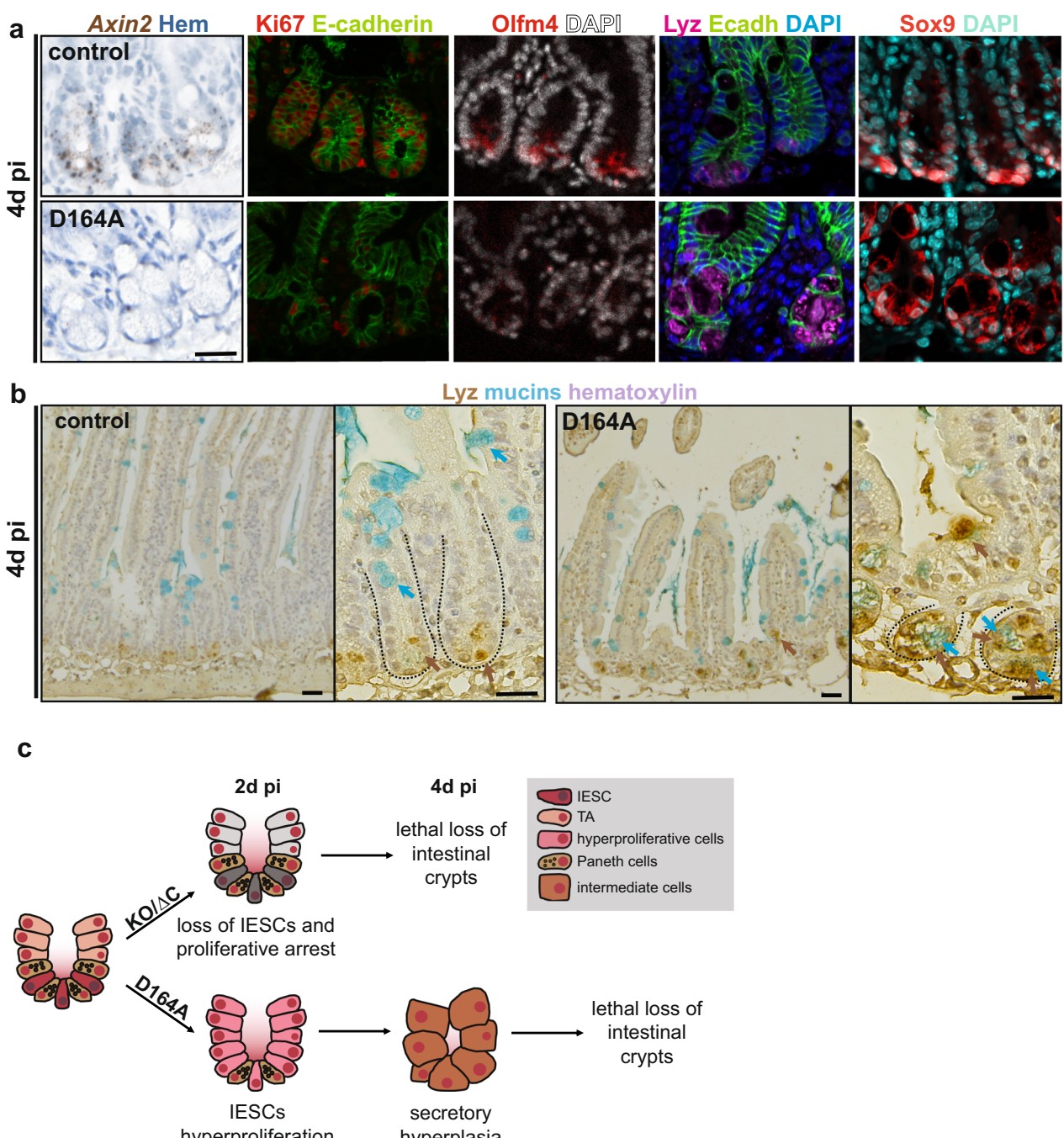

**Fig. 2 Intestinal crypts lacking the output via N-terminal recruited co-factors to exhibit stem cell loss and secretory hyperplasia 4d pi. a** *Axin2* (Wnt-target) in situ hybridization, Olfm4 (IESC marker), Ki67 (proliferating cells), lysozyme (Lyz, Paneth cells) and Sox9 (Paneth cells precursors) immunofluorescence of control (*villin-CreER^T2;Ctnnb1^wt/flox*) and D164A (*villin-CreER^T2;Ctnnb1^D164A/flox*) duodenal crypts. Hematoxylin (nuclei), DAPI (nuclei) or E-cadherin (cell shape) for counterstain. Timepoint: 4d pi. Scale bar, 20 µM. Representative images of three biological replicates. **b** Paneth cells and goblet cells visualized by Lyz-HRP and alcian blue (mucins), respectively. Hematoxylin as counterstain. Brown arrows indicate Lyz+ cells, which are also found mislocalized to the villus of D164A animals. Blue arrows indicate mucin-expressing cells. Insets show higher magnification of crypts. Timepoint: 4d pi. Scale bar, 20 µM. Representative images of three biological replicates. **c** Working model summarizing impacts of the missing contribution of N- vs. C-terminal β-catenin co-factors.

compared to controls, while their expression is lost 4d pi (Supplementary Fig. 5c, d). To disentangle the transcriptomic effects caused by impaired N-terminal interactions from those arising due to the acute loss of the conditional β-catenin allele, we devised a normalization procedure using the control time series data (Methods). The resulting normalized data thus represent the

transcriptomic changes occurring in SCEP cells, over time, upon impairing β-catenin-NTF interactions.

Confirming the data presented above, D164A SCEP cells exhibited transient upregulation of IESC and proliferation markers 2d pi (*Olfm4, Mki67, Ccnd1*), followed by increased expression of differentiation markers 4d pi (*Ada, Apoa4, Apoa1,*

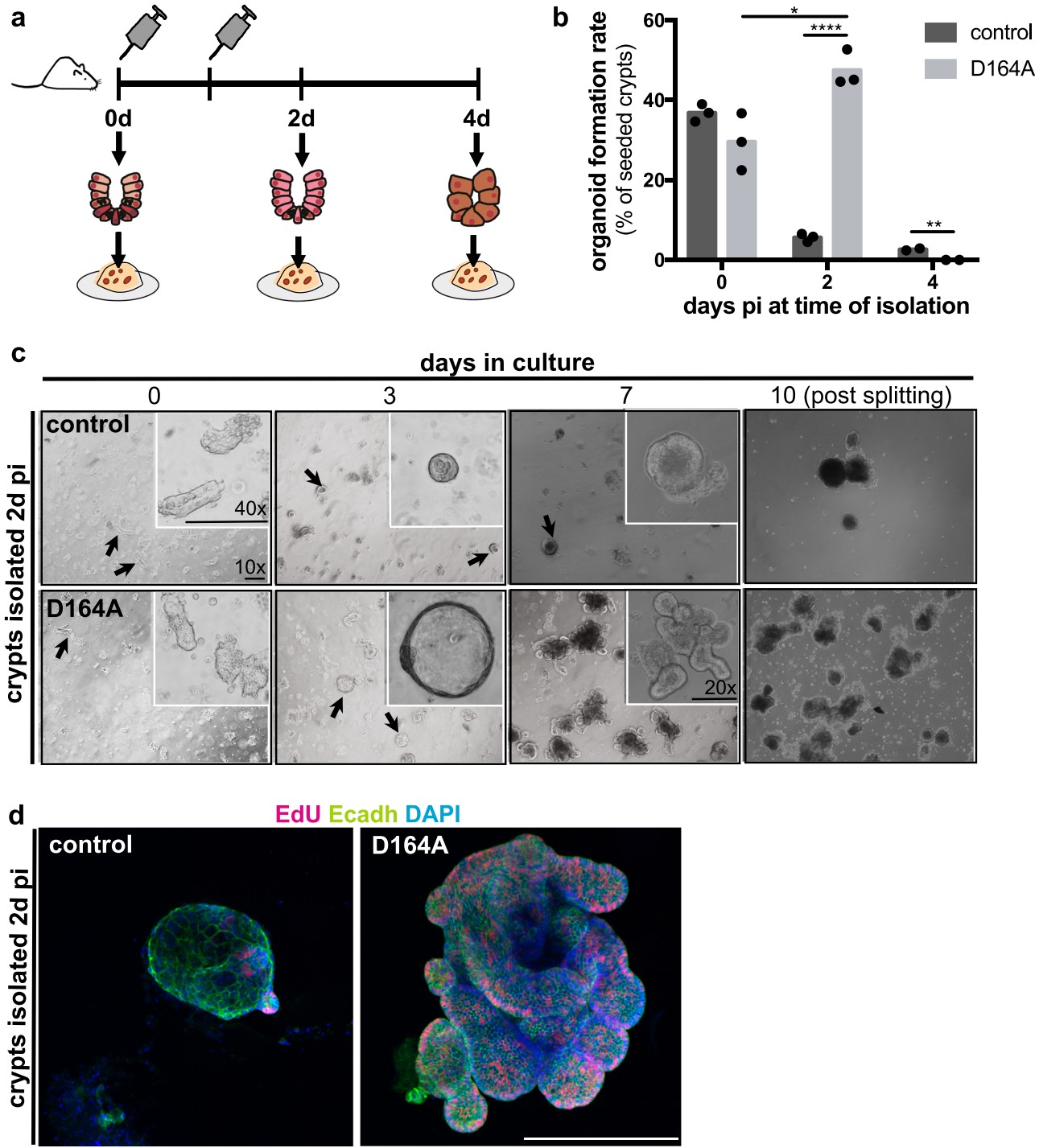

**Fig. 3 Hyperproliferative IESCs expressing only *Ctnnb1^D164A* have an increased ability to form organoids. a** Experimental design of organoid formation assay. **b** Organoid formation rate (in %) of control (*villin-CreER^T2;Ctnnb1^wt/flox*) and mutant (*villin-CreER^T2;Ctnnb1^D164A/flox*) crypts isolated 0, 2 and 4d pi. ****$p < 0.0001$, **$p = 0.0088$, *$p = 0.0207$, as calculated by unpaired, two-tailed Student's T-test ($n = 3$). Barplot indicates mean rate. **c** Brightfield images of control and mutant organoids formed from crypts isolated 2d pi grown in Matrigel for 10 days. Insets show higher magnification. Representative images of three biological replicates. Scale bars, 20 μM. **d** Confocal images of crypts isolated 2d pi and grown in Matrigel for 7 days. EdU staining shows proliferating cells. E-cadherin (cell shape) and DAPI (nuclei) as counterstain. Scale bar, 20 μM. Representative images of three biological replicates.

*Alpi, Ace1, Muc3, Lyz1*) (Supplementary Fig. 5g). This is highlighted by diffusion maps[27] as a shift along the stem cell differentiation continuum: SCEP cells isolated 2d pi have a significantly more "stem-like" distribution, whilst SCEP cells from D164A crypts 4d pi are more shifted towards differentiation (Fig. 4a, b). We also identified intermediate secretory cells in

D164A within SCEP cells 4d pi by their co-expression of Paneth and goblet cell features (Supplementary Fig. 5e).

The upregulation of stem cell markers 2d pi was accompanied by a significant increase in the mean expression levels of cell cycle-related genes[28] (Fig. 4c). Of note, this behavior of D164A SCEP cells was not observed in other cell types with

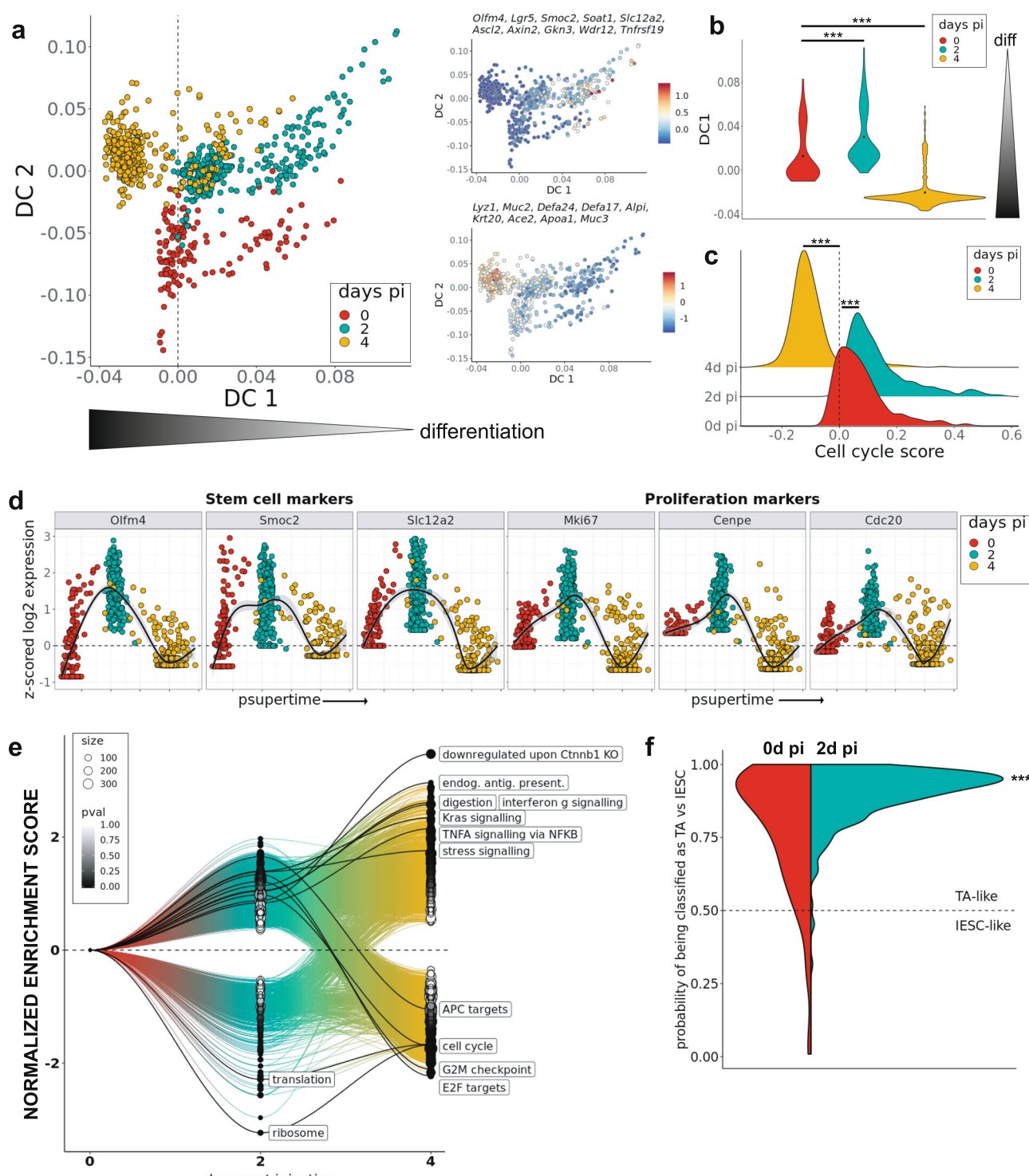

**Fig. 4 Myc-E2F-driven proliferation is followed by stress signalling and loss of stem cells in N-terminal mutant (D164A) crypts. a** Diffusion map of stem cells and early progenitors (SCEP) cells isolated from D164A (*villin-CreER^T2;Ctnnb1^D164A/flox*, *n* = 3) crypts 0, 2 and 4d pi and normalized by respective control (*villin-CreER^T2;Ctnnb1^wt/flox*, *n* = 3). Cells (dots) colored by timepoint after Cre induction (days pi). Diffusion component 1 (DC1) represents differentiation pseudotime. On the right, cells colored by their average expression of stem cell (top) and differentiation marker genes (bottom). **b** Distribution of DC1 score in normalized D164A SCEP cells over time (\*\*\**p* < 2.2e−16, two-sided Wilcoxon test). **c** Distribution of cell cycle score (average expression of cycling genes) in normalized D164A SCEPs over time (\*\*\**p* < 2.2e−16, two-sided Wilcoxon test). **d** Expression profile of stem cell and proliferation marker genes in normalized SCEP cells ordered along supervised pseudotime (psupertime). Cells (dots) colored by timepoint. **e** Gene set expression analysis (GSEA) on time-varying genes in SCEPs across timepoints. Normalized enrichment score (*y*-axis) over time (*x*-axis) of MSigDB gene sets. Dot size indicates the size of the enriched gene set. Dot color indicates *p* value of enrichment, as calculated by the hypergeometric test. Selected gene sets labeled (identifiers in the text). **f** Distribution of predicted model responses of logistic regression showing a shift of normalized SCEP cells 2d pi from intestinal epithelial stem cells (IESC) to transit amplifier (TA) traits. \*\*\**p* < 2.2e−16, as calculated by two-sided Kolmogorov-Smirnov test.

active Wnt signalling, such as in Paneth cells or enterocytes (Supplementary Fig. 5f). Supervised pseudotime analysis[29] confirmed that stem cell and proliferation markers exhibit a similar expression profile over time, increasing 2d pi, and subsequently dropping 4d pi (Fig. 4d). We had observed increased *Myc* expression in D164A crypts 2d pi (Fig. 1h), we therefore asked whether Myc-driven proliferation was inducing increased cell cycle rate of SCEP cells. Indeed, at 2d pi, Myc-dependent Wnt-targets were significantly enriched in DEGs between 0d and 2d pi (Myc-dependent APC-targets obtained from ref. [30]. MSigDB code M1757, normalized enrichment score (NES) = 1.68, $p < 0.05$) (Fig. 4e). Moreover, the signature of E2 factor (E2F)—a major cell cycle regulator, and a Myc-target—was significantly enriched (M5901, NES = 1.56, $p < 0.05$) together with genes involved in cell cycle (M14460, NES = 1.65, $p < 0.05$) and G2M checkpoint (M5901, NES = 1.72, $p < 0.05$) (Fig. 4d). These proliferation signatures were all significantly depleted 4d pi (NES $< -1$, $p < 0.05$), reflecting an abrupt proliferative arrest (Fig. 4c, d).

We wondered whether the observed Myc-E2F-induced proliferation burst induced an en-bloc conversion of D164A IESCs to TAs, which are known to exhibit higher expression of proliferative Wnt targets[31]. As our mouse models don't harbor any IESCs reporter (e.g., *Lgr5-EGFP*) that enables the identification of stem cells, we tested this hypothesis computationally. We trained a logistic regression model on TAs and IESCs obtained from a publicly available single-cell dataset of murine small intestinal epithelium[32], and tested SCEP cells from 0d and 2d pi against this model. Comparison of the distribution of predicted model responses from the two timepoints shows a significant shift of 2d pi cells towards TA traits (Fig. 4F). Cumulatively, our results indicate that impairing β-catenin-NTF interactions transiently activates an Myc-E2F-driven proliferation program in IESCs, which results in exhaustion of the self-renewing IESC pool.

**Cell-intrinsic and environmental stress signals induce mis-differentiation**. GSEA of time-varying genes in SCEP cells suggests an increase in stress signaling in crypts upon impairment of β-catenin-NTF interactions. Indeed, key mediators of the unfolded protein response (UPR)—*Xbp1*, *Creb3*, and *Creb3l3*—feature among the most strongly upregulated transcription factors in D164A SCEP cells over time (Supplementary Fig. 5h). We confirmed the ectopic expression of *Creb3l3* in D164A crypts 4d pi using smFISH (Fig. 5a). *Creb3l3* is a strongly zonated gene; its expression increases along the crypt-villus axis, with peak expression at the villus tip[33]. Moreover, expression of the UPR markers *Hspa5* and *Ddit3* in crypts 2d pi was significantly upregulated in D164A compared to controls, as independently measured by qRT-PCR (Fig. 5b). Finally, consistent with activation of UPR, genes involved in translation initiation and ribosomal function are significantly downregulated in D164A crypts (M27686, NES = $-2.52$, $p < 0.05$ and M189, NES = $-3.26$, $p < 0.001$, respectively) (Fig. 4e).

In addition to genes involved in cell-intrinsic stress, we also found cytokine-induced stress genes strongly upregulated in D164A SCEP cells over time. Indeed, signatures of Tumor Necrosis Factor α (TNFα) and Interferon γ (IFNγ) signalling were significantly enriched (M5913, NES = 2.32, $p < 0.001$ and M5890, NES = 2.08, $p < 0.001$, respectively). Moreover, the cytokine-induced genes *Irf7*, *Fos*, *Jun*, and *Junb* featured among the most upregulated transcription factors according to our supervised pseudotime analysis (Supplementary Fig. 5h). Interestingly, we found a time-dependent upregulation of genes involved in endogenous antigen presentation (M16750, NES = 3.01, $p < 0.001$), specifically genes encoding for components of the MHC class II complexes (Figs. 4e and 5c).

Our scRNAseq data thus suggest that the crosstalk between the crypt epithelial and immune compartments increases upon impairment of β-catenin-NTF interactions. We profiled the intestinal immune infiltrate by flow cytometry and found that the number of CD45$^+$ cells was more than doubled in D164A animals over time, but not in controls (Fig. 5e). Indeed, immunofluorescence analysis indicates a substantial infiltration of CD45$^+$ cells in the pericryptic zone, as well as within crypts, of N-terminal mutant animals (Fig. 5d). Specifically, leukocyte profiling revealed a significant increase of CD4$^+$ T cells, MHCII$^+$ and MHCII$^-$ monocytes, as well as neutrophils, in D164A animals. Of note, ΔC mutants show unaltered CD45$^+$ cell count and immune population composition at 2d pi (Fig. 5e and Supplementary Fig. 6a). It has recently been shown that T-cell-derived TNFα, IFNγ and interleukin 17 (IL-17) inhibit IESC renewal[34], and that excess interferon-γ signalling induces mis-differentiation towards the secretory fate[35]. Upon restimulation with PMA and ionomycin, we observed a substantial, albeit not significant, increase in TNFα, IFNγ and IL-17 secretion from CD4$^+$ and CD8$^+$ T cells isolated from D164A animals 4d pi (Supplementary Fig. 6b).

We asked whether the influx of immune cells was caused by a breach of the intestinal epithelial barrier, rather than crypt-derived signals. To assay intestinal permeability, we administered fluorescein isothiocyanate-conjugated dextran (FITC-dextran) by oral gavage to ΔC, D164A and control mice 2d and 4d pi. Consistent with our previous observations indicating intact epithelial integrity, we found no significant difference ($p > 0.01$) in FITC-dextran concentration in the plasma of mutant animals (Fig. 5g).

Altogether, these results suggest that, following their hyper-proliferation induced by impaired β-catenin-NTF interactions, IESCs are subjected to combination of cell-intrinsic (UPR) and environmental (cytokines) stresses, which inhibit their self renewal[34–36]. Notably, antigen presentation is intrinsically linked to UPR, as *Xbp1* is also a regulator of MHC class II genes[37,38]. Loss of stem cells and aberrant differentiation of the crypt, accompanied by increasing immune infiltration, inevitably result in failure of epithelial homeostasis.

**The chromatin landscape of D164A crypts reflects the transition to proliferative arrest and mis-differentiation**. Our analyses revealed rapid and profound transcriptomic and morphological changes in the crypts upon impairment of β-catenin-NTF interactions (D164A mutant). We reasoned that profiling the chromatin landscape of D164A crypts would provide insight into the gene regulatory mechanisms responsible for such drastic reprogramming of epithelial cells. To this end, we performed Assay for Transposase-Accessible Chromatin using sequencing (ATACSeq) of control and D164A crypts isolated 2d pi. Differential peak analysis revealed that 1469 ATAC-peaks are differentially accessible (logFC > |1|& $p < 0.01$) in D164A crypts, compared to control crypts (Fig. 6a, b). We subjected these peaks to motif analysis with HOMER[39], and found a significant enrichment (Benjamini-corrected $q$-value < 0.001) of motifs associated with transcription factors involved in intestinal differentiation, such as hepatocyte nuclear factors (HNF4a,1,1b), ETS-related transcription factors (EHF, ELF3,5), Caudal Type Homeobox (Cdx2,4), GATA-binding factors and Krüppel-like factors (Supplementary Fig. 7a). These findings correlate with increased expression of secretory and absorptive lineage markers in D164A crypts seen at 4d pi. Moreover, peaks annotated to Wnt target genes

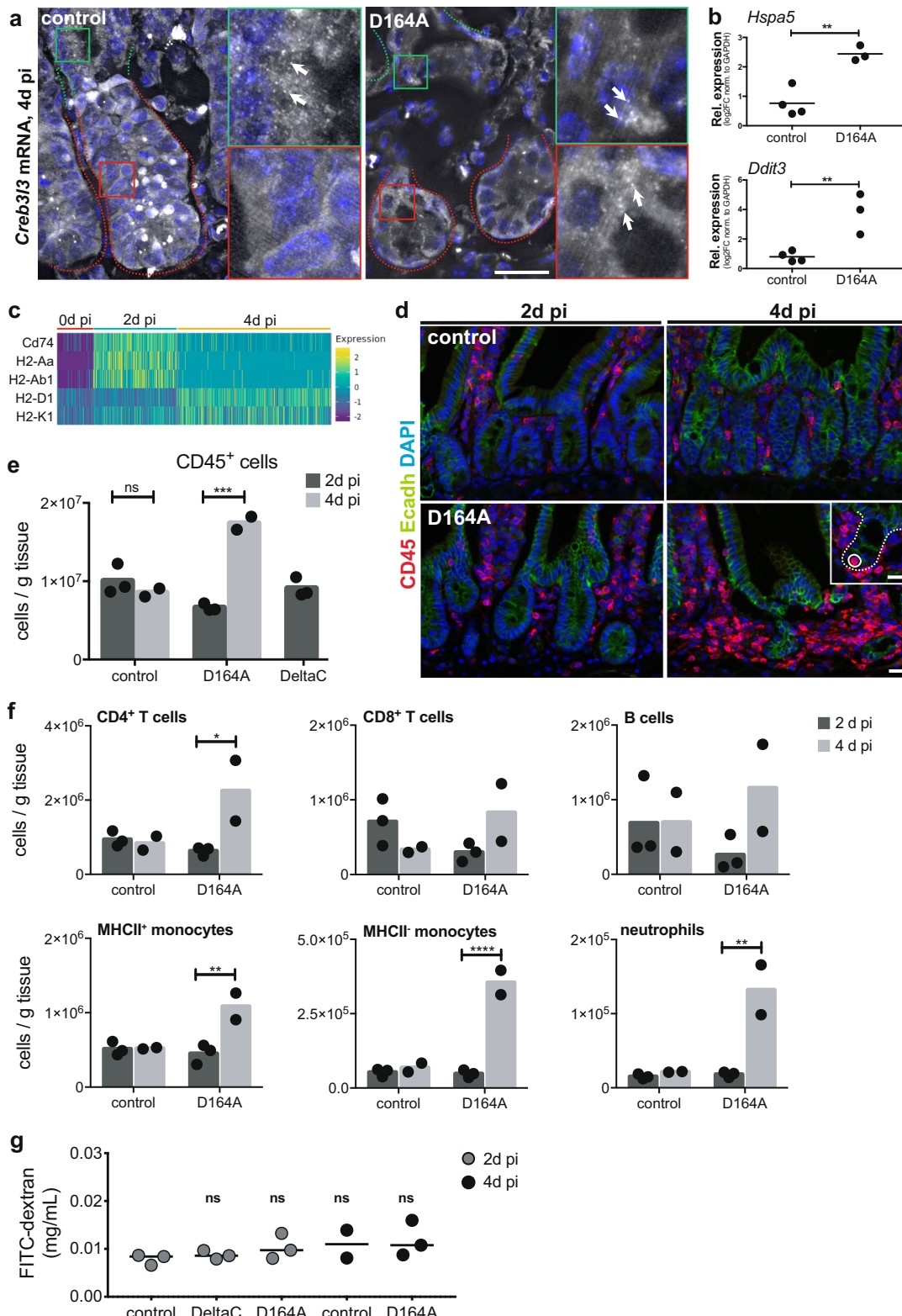

(*Tcf7l1/2, Axin2, Fzd7, Sp5*), Wnt-inhibitors (*Tle3, Trabd2b, Wif1, Dab2ip, Amer3*), and cell cycle genes (*E2f1, Cenpf, Cebpe*) figure among the lost peaks (Fig. 6b). These results indicate that the chromatin landscape 2d pi preludes the proliferative arrest and secretory hyperplasia observed in N-terminal mutants (D164A) 4d pi. Indeed, we found that a significant fraction of gene sets overrepresented in D164A vs control crypts according to our ATAC-peak analysis (2d pi) were also

overrepresented in the transcriptome of D164A mutant crypts (4d pi) (Supplementary Fig. 7b).

**JNK signalling mediates major chromatin remodeling in D164A crypts**. In line with our scRNAseq results the analysis of the ATACSeq data revealed a strong enrichment of motifs bound by transcription factors downstream of the c-Jun N-terminal

**Fig. 5 UPR stress and immune infiltration in D164A crypts 4d pi. a** smFISH shows villus-restricted localization of *Creb3l3* mRNA in control animals (*villin-CreER^T2;Ctnnb1^wt/flox*) and its ectopic expression in D164A (*villin-CreER^T2;Ctnnb1^D164A/flox*) crypts. Red and green dotted lines indicate crypt and villus area, respectively. Red and green insets show higher magnification of crypt and villus area, respectively. Arrows indicate single mRNA molecules visible as dots. Timepoint: 4d pi. Scale bar, 20 μM. Representative images of three biological replicates. **b** qRT-PCR indicates increased expression of UPR *Hspa5* (**$p = 0.0028$) and *Ddit3* (**$p = 0.0077$) markers in D164A crypts ($n = 3$) with respect to control ($n = 4$). Expression levels normalized to GAPDH. Timepoint: 4d pi. Unpaired, two-tailed Student's *T*-test. **c** Increased expression of genes encoding for components of MHC class II complexes in normalized D164A SCEP cells over time. **d** CD45 immunofluorescence in control and D164A small intestinal sections 2 and 4d pi. Inset show immune cell (circled) infiltration within crypt (dashed line). Scale bar, 20 μM. Representative images of three biological replicates. **e** Total leukocyte counts per mg of tissue in control, D164A and ΔC (*villin-CreER^T2;Ctnnb1^ΔC/flox*) animals 2d pi ($n = 3$) and 4d pi ($n = 3$). ***$p = 0.0004$, as calculated by one-way ANOVA with Tukey's post-test. Barplot indicates mean counts. **f** Counts per mg of tissue of CD4⁺ (*$p = 0.0394$) and CD8⁺ T-cell, B-cells, MHCII⁺ (**$p = 0.0066$) and MHCII⁻ monocytes (****$p < 0.0001$), and neutrophils (**$p = 0.001$) 2d pi ($n = 3$) and 4d pi ($n = 3$). One-way ANOVA with Tukey's post-test. Barplot indicates mean counts. **g** Intestinal permeability of control, D164A and ΔC animals 2d and 4d pi, as quantified by FITC-dextran concentration (mg/mL) in plasma. Horizontal line indicates mean concentration. ns $= p > 0.01$, as calculated by one-way ANOVA with Tukey's post-test ($n = 3$, except for control 4d pi where $n = 2$).

kinases (JNK) signalling, namely the AP-1 transcription factors Fra1/2, Fos, Fosl2, JunB, Atf3, and BATF (Fig. 6c). Cumulatively, these motifs were enriched in 220 out of the 600 gained peaks in D164A (37%), including peaks mapped to *Fos*, as well as *Mapk8* (JNK) and the closely related *Mapk13* (SAPK4), indicating robust JNK-mediated activation of gene expression (Fig. 6d and Supplementary Fig. 7a). We validated JNK pathway activation by immunofluorescence, which reveals strong nuclear accumulation of phosphorylated JNK (pJNK) in D164A, but not in control crypts 4d pi (Fig. 6e). Similar to *Creb3l3*, the expression of transcription factors *Fos* and *JunB* is normally restricted to enterocytes at the villus tip[33,40]. However, in D164A animals 4d pi, *Fos*, and *JunB* transcripts are ectopically expressed in crypts (Fig. 6f), corroborating the ATAC-Seq and scRNASeq results. Moreover, the transmembrane mucin glycoprotein *Muc13*, which was independently shown to be induced by JNK activity[41], also figure among the gained peaks in D164A mutant crypts (Fig. 6b), in line with increased alcian blue positivity of hyperplastic and mis-differentiated intermediate cells 4d pi. Altogether, these results indicate that, following lack of β-catenin-NTF interactions, JNK signalling is activated. While mis-differentiation of IESCs upon perturbation of niche signalling has been previously described, the JNK pathway hasn't, so far, been implicated in the loss of IESCs.

**Bcl9/9L are essential upon acute reduction in β-catenin levels.** We wondered why the sole presence of β-catenin-D164A leads to stem cell exhaustion, while the loss of BCL9/9L only transiently downregulates IESC markers[13–16]. Possibly, this discrepancy is due to the fact that, in our model, the impairment of β-catenin-NTF interactions occurs concomitantly with an acute reduction in β-catenin levels (recombination of the floxed wt allele). Indeed, while dispensable for intestinal epithelial homeostasis, BCL9/9L are required when homeostasis is challenged, e.g., upon deregulation of Wnt signalling by LGK974 inhibition[15], DSS treatment[13] or irradiation[15]. Supporting this notion, simultaneous deletion of BCL9/9 L and one copy of β-catenin in *villin-CreER^T2; Ctnnb1^wt/flox;Bcl9^flox/flox;Bcl9l^flox/flox* animals, leads to humane endpoint 6d pi (Fig. 7a). Morphologically, we observed crypt atrophy, mesenchymal thickening, and villus shortening, as well as crypts reminiscent of the secretory hyperplasia observed in D164A animals (Fig. 7b). In the presence of two copies of β-catenin, loss of BCL9 and BCL9L leaves the intestinal morphology unaltered (Fig. 7b).

Cumulatively, our data indicate that transcriptional co-factors selectively modulate β-catenin-mediated Wnt-target gene expression. We propose a model in which C-terminal co-activators govern basal Wnt signaling levels. Impairing their activity leads to loss of stem cells and proliferative arrest. In contrast, N-terminal co-factors maintain stem cell state. Blocking their contribution

results in hyperproliferation and aberrant differentiation of IESCs (Fig. 7c).

## Discussion

We have previously shown that the C- and N-terminal transcriptional outputs of Wnt/β-catenin signalling have distinct and independent functions during embryonic development[8]. In this study, we aimed to uncover their specific contributions to the maintenance of adult IESC homeostasis. We show that the cellular dynamics of crypt collapse upon impairment of β-catenin-BCL9/9L interactions are profoundly distinct from the rapid crypt atrophy observed upon β-catenin deletion or impairment of interactions with C-terminally recruited co-factors. Our data suggest that C-terminally recruited co-activators act as a binary on-off-switch of transcription, and are therefore essential for the basal β-catenin-mediated transcription of Wnt-target genes. Conversely, N-terminal co-factors fine-tune β-catenin's transcriptional output to the levels required for proper proliferation and self-renewal of IESCs. The results presented herein provide evidence that differential regulation of β-catenin transcriptional outputs preserves the narrow window of Wnt-pathway activity required to govern the identity and fate of IESCs. This selective transcriptional modulation may be key in preserving "just-right" levels of Wnt signaling, which have been implicated in IESC homeostasis both physiological and in tumor context[42].

Compelling evidence has recently been provided that the intestinal IESC niche is readily regenerated upon perturbations, as differentiated cells can revert to stem cells[43–46]. Epithelial plasticity can thus mask the effects of loss of function studies of proteins, such as BCL9/9L, whose contribution to intestinal homeostasis only becomes apparent when the epithelial stem cell niche is challenged. While increasing attention has been devoted to the role of these proteins in maintaining colorectal cancer stemness, the function of β-catenin-NTF-interactions in healthy intestinal homeostasis has been overlooked. Our data suggest that N-terminal co-factors selectively restrain Wnt/β-catenin outputs that promote Myc-E2F-driven proliferation, thereby ensuring the maintenance of a self-renewing pool of IESCs. Indeed, in N-terminal mutant crypts, excessive proliferation of IESCs rapidly led to the exhaustion of the stem cell pool. Concomitantly with robust JNK pathway activation, the crypts undergo a profound mis-differentiation and "villisation". Indeed, we show ectopic crypt expression of gene programs characteristic of enterocytes at the villus tip[33,40], where terminally differentiated cells are eliminated by apoptosis. Mounting inflammation and colitis eventually result in a lethal loss of intestinal function. Interestingly, a proliferative subset of IESCs has recently been described to function as unconventional antigen-presenting cells, orchestrating interactions with immune cells in the intestine via expression of MHC class II complexes[34]. On the other hand, T-cell-derived signals

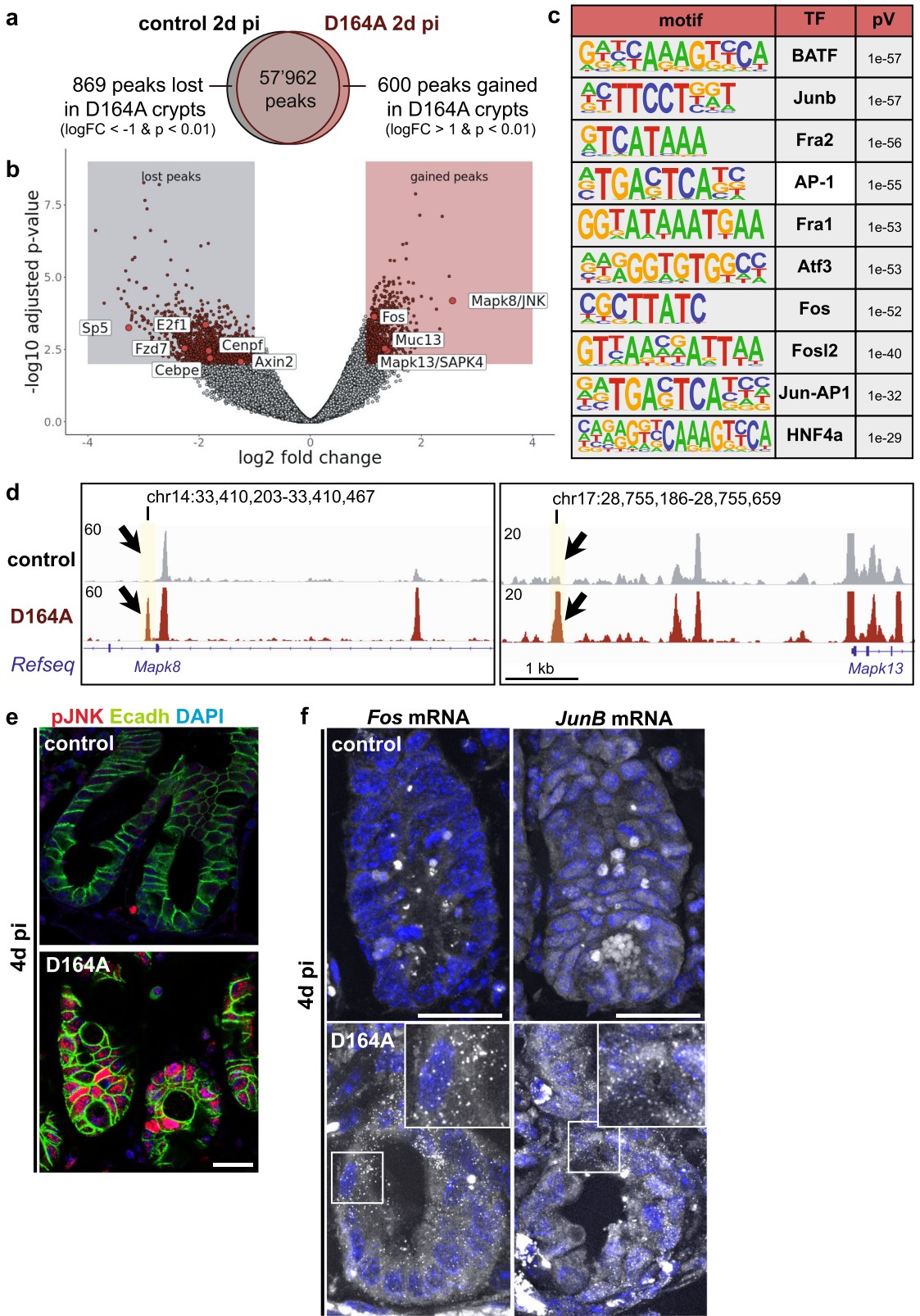

are able to inhibit self-renewal and induce differentiation of IESCs[34–36]. It is tempting to speculate that cytokine-induced terminal differentiation of stem cells, possibly via the JNK pathway, is an in-built mechanism to prevent their abnormal proliferation. However, further studies should dissect the role of immune cell-derived signals for the maintenance of intestinal homeostasis in both physiological and perturbed conditions.

## Methods

**Mice.** The *Ctnnb1-flox* allele[47] was combined with the *Ctnnb1-delC, Ctnnb1-dm* or *Ctnnb1-D164A*[8], and bred into the VillinCre-ER^T2^ mouse strain (The Jackson Laboratory) to generate *villin-CreER^T2^;Ctnnb1^wt/flox^* (control) *villin-CreER^T2^; Ctnnb1^flox/flox^* (KO), *villin-CreER^T2^;Ctnnb1^dm/flox^* (dm), *villin-CreER^T2^; Ctnnb1^D164A/flox^* (D164A) and *villin-CreER^T2^; Ctnnb1^ΔC/flox^* (ΔC). In these mice, the conditional β-catenin allele can be deleted specifically in the intestinal epithelium by tamoxifen-inducible VillinCre-ER^T2^-mediated recombination, leaving

**Fig. 6 JNK signalling mediates major chromatin remodeling and triggers the expression of villus tip genes in D164A crypts. a** Venn diagram of measured ATACseq peaks in crypts isolated from control (*villin-CreER^T2;Ctnnb1^wt/flox*, n = 3) and D164A (*villin-CreER^T2;Ctnnb1^D164A/flox*, n = 3) animals. Timepoint: 2d pi. Differential peak analysis indicates that 600 peaks are gained (logFC >1 & p < 0.01) and 869 are lost (logFC < −1 & p < 0.01, as calculated by edgeR's exact test) in D164A crypts. **b** Volcano plot of ATACseq peaks in control and mutant crypts. Gained peaks (logFC >1 & p < 0.01) evidenced in red box. Lost peaks (logFC < −1 & p < 0.01) evidenced in gray box. Selected peaks are labeled. *P*-values calculated with edgeR's exact test. **c** Table of the 10 most significantly enriched transcription factor (TF) binding motifs in differential peaks obtained from HOMER motif analysis (hypergeometric test). **d** Combined ATACSeq tracks of control and D164A counts. Black arrows show gained peaks with AP-1 binding motif annotated to Mapk8/JNK and Mapk13/SAP1. **e** Immunostaining of phospho-JNK in control and D164A duodenal sections. E-cadherin (cell shape) as counterstain. Timepoint: 4d pi. Scale bar, 20 μM. Representative images of 3 biological replicates. **f** smFISH of *Fos* and *JunB* mRNA in control and D164A duodenal sections. DAPI (nuclei) as counterstain. Insets show higher magnification. Arrows indicate single mRNA molecules visible as dots. Timepoint: 4d pi. Scale bar, 20 μM. Representative images of three biological replicates.

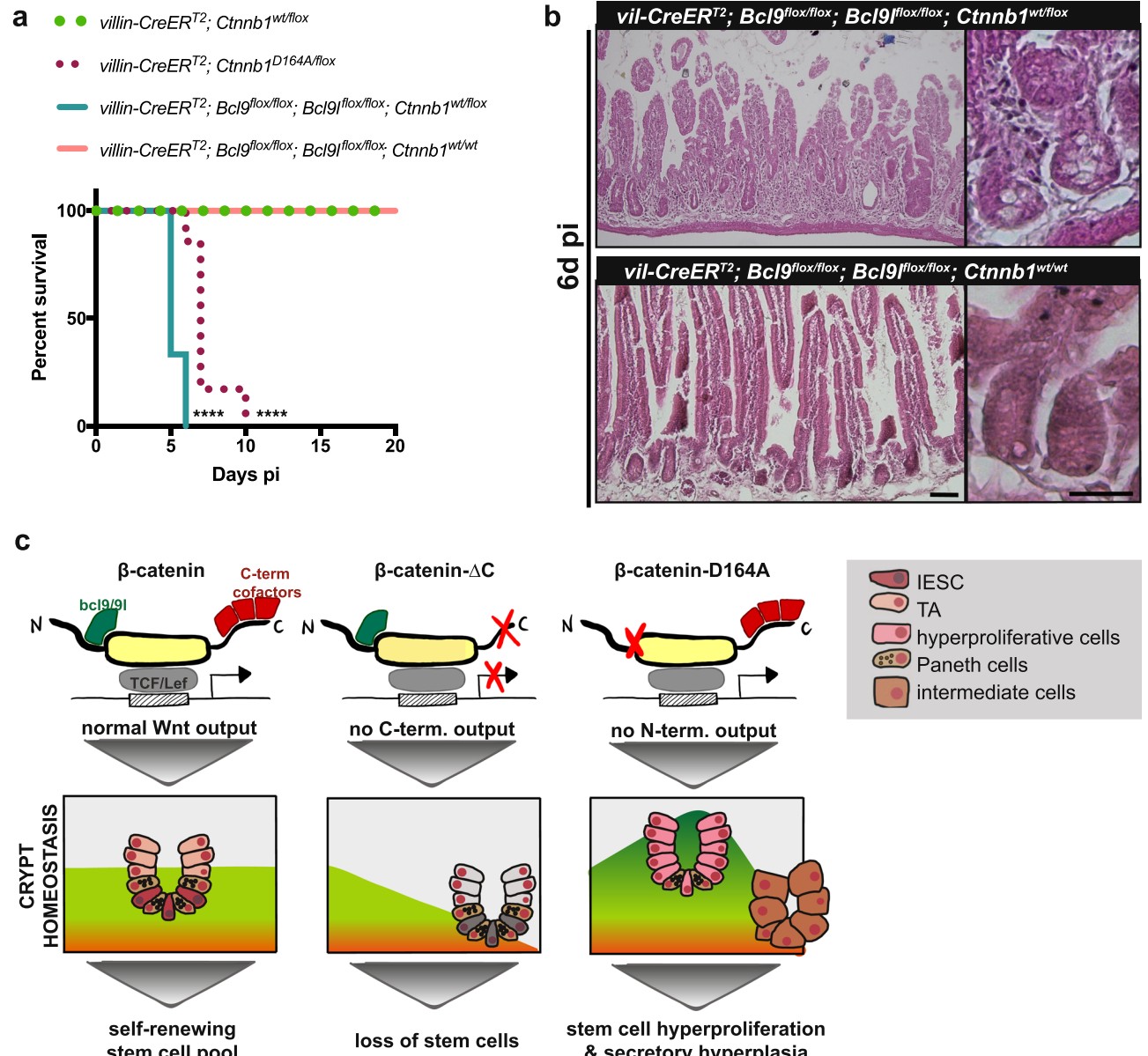

**Fig. 7 Selective regulation β-catenin transcriptional outputs govern the identity of the intestinal epithelial stem cells. a** Survival plot of *villin-CreER^T2; Ctnnb1^wt/flox* (control, n = 3), *villin-CreER^T2;Ctnnb1^D164A/flox* (D164A, n = 5), *villin-CreER^T2;Bcl9^flox/flox;Bcl9l^flox/flox;Ctnnb1^wt/flox* (n = 3) and *villin-CreER^T2; Bcl9^flox/flox;Bcl9l^flox/flox;Ctnnb1^wt/wt* animals (n = 3). ****p < 0.0001 as calculated by two-sided log-rank (Mantel-Cox) test. **b** Histology sections of *villin-CreER^T2;Bcl9^flox/flox;Bcl9l^flox/flox;Ctnnb1^wt/flox* and *villin-CreER^T2;Bcl9^flox/flox;Bcl9l^flox/flox;Ctnnb1^wt/wt* animals. Concomitant deletion of Bcl9, Bcl9L, and one copy of β-catenin induces crypt atrophy and villus shortening. Timepoint: 6d pi. Scale bar, 20 μM. Representative images of three biological replicates. **c** Schematics of the proposed model. C-terminal co-activators govern basal Wnt signaling and preventing their activity leads to loss of stem cells and proliferative arrest. Contrary, N-terminal co-factors act as stem cell specifiers. Blocking their contribution results in hyperproliferation and aberrant differentiation of intestinal epithelial stem cells. IESC intestinal epithelial stem cell, TA transit amplifier cell.

Wnt/β-catenin-mediated transcription under the sole control of mutant β-catenin (wt in control animals). The *Ctnnb1-D164A* allele was detected with the primers 5′-TCCCTGAGACGCTAGATG-3′ and 5′-GAGTCCCAGCAGTACAAC-3′, yielding an amplicon of the size 475 bp for the wild-type and 628 bp for mutant alleles[8]. *Ctnnb1-ΔC* allele was determined using primers 5′-GTGCACACGTCATGCTTTA C-3′ and 5′-TGGCTTGTCCTCAGACATTCG-3′, which generate an amplicon of size 349 bp for the wild-type and 415 bp for mutant alleles[8]. The presence of the *villin-CreER^T2* allele was detected with the primers 5′-CAAGCCTGGCTCGACG GCC-3′ and 5′-CGCGAACATCTTCAGGTTCT-3′, which generate a 220 bp product[17]. To induce Cre-ER^T2-mediated recombination, tamoxifen (Sigma, 80 mg/kg) was injected intraperitoneally on two consecutive days. Recombination of the conditional β-catenin allele was confirmed for every mouse included in this study by PCR with the primers 5′-AAGGTAGAGTGATGAAAGTTGTT-3′ (RM41), 5′-CACCATGTCCTCTGTCTATTC-3′ (RM42) and 5′-TACACTATTGAATCACAG GGACTT-3′ (RM43), generating products of 221 bp for the wild-type allele, 324 bp for the floxed allele and 500 bp for the floxdel allele[47] (Supplementary Fig. 1d). To eliminate Bcl9 and Bcl9L, previously published[13] conditional alleles were combined with the *villin-Cre-ER^T2* allele (*villin-CreER^T2;Bcl9^flox/flox, Bcl9l^flox/flox*) and with the *Ctnnb1-flox* allele (*villin-CreER^T2;Bcl9^flox/flox; Bcl9l^flox/flox; Ctnnb1^flox/wt*). All genotyping primer sequences can be found in Supplementary Table 1.

Mice were sacrificed 48 h after the first tamoxifen injection. Alternatively, mice were injected on 4 consecutive days and sacrificed 96 h after the first injection, or injected on 5 consecutive days and sacrificed 6 or 7 days after the first injection. Mouse experiments were performed in accordance with Swiss guidelines and approved by the Veterinary Office of the Kanton of Zurich, Switzerland. All animals were kept on a C57BL/6 background. Mice were 8–12 weeks old at the time of treatments and cell isolations. Mice of both sexes were used in all experiments and littermates were used as controls.

**Immunohistochemistry.** Dissected duodenum was flushed thoroughly in ice-cold PBS, then cut in 1 cm pieces and fixed in 4% PFA in PBS overnight at 4 °C. After repeated washing in PBS, tissues were dehydrated in a spin tissue processor, embedded in paraffin, and cut in 8 μm sections. Deparaffinized tissue sections were subjected to antigen retrieval in 2.4 mM sodium citrate and 1.6 mM citric acid, pH 6, for 25 min in a steamer. Sections were washed with PBST (0.1% Tween-20 in PBS) and blocked for 30 min at RT in blocking buffer (5% BSA, 5% heat-inactivated normal goat serum in PBST). Following overnight incubation at 4 °C with primary antibody (1:100, in blocking buffer, Supplementary Table 1), sections were washed in PBST and incubated with secondary antibody (1:400, in blocking buffer) for 1 hr at room temperature. Nuclei were stained with DAPI (Sigma, 1:1000) in blocking solution for 5 min at RT. Sections were imaged on a Leica LSM 710 confocal microscope and processed equally using the ImageJ (FIJI) software, or scanned on Vectra 3.0 Automated Quantitative Pathology Imaging System (Perkin Elmer). The percentage of Ki67+ crypt cells was automatically quantified with the software inForm Cell Analysis software (Perkin Elmer). Crypt area was manually defined. We quantified at least 30 crypts from 2 to 3 different mice per condition. Barplots were generated on GraphPad Prism. The unpaired Student's *T* test function in GraphPad Prism was used to analyze the significance of two-group comparisons.

**Double histology for intermediate cells.** Deparaffinized tissue sections were incubated with hydrogen peroxide for 5 min to block endogenous peroxidase, then incubated for 30 min in blocking buffer (5% BSA, 5% heat-inactivated normal goat serum in PBST) for 30 min. After overnight incubation with primary anti-Lyz antibody (Dako), sections were repeatedly washed in PBST and incubated with biotinylated secondary antibody for 1 h at RT and then visualized with VEC-TASTAIN ABC HRP Kit (Vector Laboratories) as indicated by the manufacturer. Sections were then incubated in a 3% aqueous solution of acetic acid for 3 min and then mucins were stained with alcian blue (1 g in 100 mL 3% acetic acid) for 15 min. Hematoxylin (50% aqueous solution) staining was then carried out according to standard procedure. Sections were imaged on Leica THUNDER 3D Live Cell Imaging system.

**RNA in situ hybridization.** Duodenum was flushed thoroughly in ice-cold PBS, then cut in 1 cm pieces and fixed under gentle agitation in 10% formalin for 20 h at room temperature. *Axin2* mRNA in situ hybridization was performed with the ACD RNAscope kit (ACDBio), according to the manufacturer's instructions. Sections were imaged on a Vectra 3.0 Automated Quantitative Pathology Imaging System and quantification was performed with the inForm Cell Analysis software (Perkin Elmer). The percentage of *Axin2* + RNA crypt cells was automatically quantified as for Ki67 staining.

**Single molecule in situ hybridization.** Mice were sacrificed and the duodenum was removed and flushed thoroughly in ice-cold PBS. Duodenal tissue was then cut in 1 cm pieces and fixed in 4% PFA (Santa Cruz Biotechnology, sc-281692) in PBS for 3 h and subsequently agitated in 30% sucrose, 4% PFA in PBS overnight at 4 °C. Fixed tissues were embedded in TIssue-Tek OCT Compound (Sakura, 4583). Eight micrometers thick sections were sectioned onto poly L-lysine coated coverslips. Probe libraries were designed using the Stellaris FISH Probe Designer Software

(Biosearch Technologies) (see Supplementary Data 1) and coupled to Cy5 (*Myc, Fos, Creb3l3*) or TMR (*Lgr5, JunB*) as described[48]. The intestinal sections were hybridized with smFISH probe sets according to a previously published protocol[49]. DAPI (Sigma-Aldrich) was used as a nuclear counterstain. smFISH imaging was performed on a Leica THUNDER 3D Live Cell Imaging system, using the following THUNDER Computational Clearing Settings in with the LAS-X software: Feature Scale (nm): 350, Strength (%): 98, Deconvolution settings: Auto and Optimization: High. The of Lgr5+ area per crypt was manually quantified using FIJI. We quantified at least 8 crypts from 2 to 3 different mice per condition. Barplots were generated on GraphPad Prism. The unpaired Student's *T* test function in GraphPad Prism was used to analyze the significance of two-group comparisons.

**Crypt isolation and culture.** Duodenal crypts were isolated and cultured as described[50] with minor modifications. The duodenum was dissected out, flushed thoroughly, opened longitudinally, and then cut in 1–2 mm stripes. After repeated washing in ice-cold PBS, the duodenal fragments were incubated for 20 min in Gentle Cell Dissociation Reagent (Stemcell Technologies). Crypts were gradually released from the tissue during four rounds of washing in 0.2% BSA and filtering through a 70 micron filter, generating 4 fractions. Only fractions 3 and 4 containing intestinal crypts were used for downstream experiments (organoids or single-cell suspension). Crypts were seeded in 50 μL Matrigel (Corning) domes and cultured in Intesticult (Stemcell Technologies). Upon passaging, the culture medium was supplemented with 1 μM Rho-associated protein kinase (ROCK) inhibitor (Y27632, Millipore). Recombination was induced with 100 nM 4-hydroxytamoxifen (4OHT, Sigma) added to the culture medium upon splitting. Wnt3a-conditioned medium was obtained as described[51].

For the organoid formation assay, mice have sacrificed 0, 2 or 4d post injection. Crypts were seeded in equal amounts (500 crypts) in 50 μL Matrigel domes. Organoids were quantified manually 7 day post crypt seeding and micrographs were collected using the AmScope 5.2. software. For EdU staining, crypts were seeded on glass-bottomed 8-well chambered slides (Thermo Fisher Scientific, Lab-TekTM, 154532) in 20μL drops. EdU incorporation (30 min pulse) and visualization were performed according to the manufacturer's instruction with Edu Click 647 Kit (baseclick) prior to overnight incubation with anti-Ecadherin (BS Transduction Lab) antibody for cell shape staining. Imaging was performed on a Visitron CSU-W1 spinning disk confocal microscope with a CFI PlanFluor ×20 objective.

**Western blotting.** Small intestinal crypts were isolated as described above. To ensure efficient lysis, crypt cells were dissociated into single cells by 5 min incubation in TripLE. After neutralization with FBS, cells were washed with PBS, resuspended in 200 μL hypotonic buffer (10 mM Hepes pH 7.9, 1.5 mM MgCl₂, 10 mM KCl, 0.5 mM DTT, cOmplete ULTRA Protease Inhibitor Cocktail (Sigma)) and incubated 10 min on ice. 10% NP-40 (50 μL) was added prior to vortexing and 1 min centrifugation at 7000 × *g*. The supernatant (cytosolic extract) was removed, sonicated for 5 min, and stored at −80 °C. The nuclear pellet was resuspended in 200 μL hypertonic buffer (20 mM Hepes pH 7.9, 1.5 mM MgCl₂, 0.42 M NaCl, 0.5 mM DTT, 0.2 mM EDTA, 25% glycerol, cOmplete ULTRA Protease Inhibitor Cocktail (Sigma)) and rotated at 4 °C for 20 min. After 5 min centrifugation at max speed, the supernatant (cytosolic extract) was removed, sonicated for 5 min, and stored at −80 °C. Protein concentration was determined with the BCA assay. Protein lysates were boiled with NuPAGE Sample Denaturing Agent (4×) and Reducing Agent (10×) (Invitrogen), separated on NuPAGE 4–12% Bis-Tris gels and transferred on nitrocellulose membranes. HRP-conjugated secondary antibodies bound to monoclonal anti-β-catenin (Novus NBP1-32239), mouse monoclonal anti-β-actin (Santa Cruz Biotechnology, sc-47778), and mouse monoclonal anti-lamin A/C (eBioscience, 14-9847-82) were visualized with WesternBright Quantum HRP substrate (advansta). Uncropped blots are found in the source data file.

**Quantitative real-time PCR.** Crypt RNA was isolated by phenol-chloroform extraction and reverse transcribed with the cDNA synthesis kit (Takara Bio Inc.) according to the manufacturer's instructions. One nanogram of total RNA was used. Expression of genes of interest was quantified by qRT-PCR using the Applied Biosystems SYBR Green Kit monitored by the QuantStudio3 system (Applied Biosystems). Samples were measured in technical triplicate and average cycle threshold values were normalized to GAPDH using the ΔΔCT method[52].

**FITC-dextran intestinal permeability assay.** Control ($n = 5$), ΔC ($n = 3$), and D164A ($n = 6$) mice were orally administered with 0.6 mg per g body weight of 4kD fluorescein isothiocyanate-conjugated dextran (FITC-dextran) (Sigma) by oral gavage 2d or 4d pi. Food and water were withdrawn. Mice were sacrificed 4 h after FITC-dextran administration, ca 1 mL blood was drawn post mortem and collected in BD Microtainer SST Tubes. The blood was centrifuged at 5000 × *g* for 10 min. Plasma was diluted with the same volume of PBS and analyzed for FITC concentration at an excitation wavelength of 490 nm and an emission wavelength of 520 nm on a TECAN Infinite 200Pro instrument.

**Leukocyte isolation for flow cytometry**. Control ($n = 5$), ΔC ($n = 3$) and D164A ($n = 5$) mice were sacrificed 2d or 4d pi. The duodenum was open longitudinally, washed and cut into pieces. Peyer's patches were removed. Pieces were weighted and incubated in HBSS with 10% FCS, 100 U/mL penicillin/streptomycin and 5 mM EDTA at 37 °C in a shaking incubator. Tissues were then digested at 37 °C for 50 min with 15 mM Hepes, an equal mixture of 250 U/mL type IV and type VIII collagenase (Sigma-Aldrich), and 0.05 mg/ml DNase I in RPMI-1640 medium supplemented with 10% FBS and 100 U/ml penicillin/streptomycin. Cells were then layered onto a 40/80% Percoll gradient, centrifuged, and the interface was washed in PBS. Total leukocyte counts were determined by adding countBright Absolute Counting Beads (Life Technologies) to each sample before flow cytometry for normalization to tissue weight. For surface staining, cells were stained in PBS with 0.5% BSA with the fixable viability dye eFluor 780 (1:1000, eBioscience) and a combination of the following antibodies (1:200): anti-mouse B220 (RA3-6B2), CD11b (M1/70), CD11c (N418), CD4 (RM4-5), CD45 (30-F11), F4/80 (BM8), Ly6G (1A8), Ly6C (HK1.4), CD103 (2E7), CD8 (53–6.7), F4/80 (MB8), MHC-II (M5/114.15.2), TCR-β (H57-597), all from BioLegend. Fc block (anti-CD16/CD32, Affymetrix) was included to minimize nonspecific antibody binding. For intracellular cytokine staining of T cells, cells were incubated for 3.15 h in complete IMDM containing 0.1 μM PMA and 1 μM ionomycin with 1:1000 brefeldin A (eBioscience) and GolgiStop solutions (BD Biosciences) at 37 °C in a humidified incubator with 5% CO$_2$. Following surface staining, cells were fixed and permeabilized with the Cytofix/Cytoperm Fixation/Permeabilization Solution Kit (BD Biosciences) according to the manufacturer's instructions. Cells were then stained for 50 min with antibodies to IL-17A (TC11-18H10.1), IFN-γ (XMG1.2), and TNF-α (MP6-XT22) all from Biolegend. Samples were acquired on a LSRII Fortessa (BD Biosciences) and analyzed using FlowJo software v10.6.2 (Becton Dickinson & Company).

**RNA sequencing**. Control ($n = 3$), KO ($n = 2$), dm ($n = 3$), ΔC ($n = 2$), and D164A ($n = 3$) mice were sacrificed 2d pi and the duodenal epithelial RNA isolation was performed as described[53]. Briefly, dissected duodenum was flushed thoroughly in ice-cold PBS, then opened longitudinally and incubated for 20 min on ice in dissociation buffer 1 (30 mM EDTA, 1.5 mM DTT in PBS), followed by 10 min incubation in dissociation buffer 2 (30 mM EDTA in PBS) at 37 °C under gentle agitation. The epithelium was released by vortexing, pelleted by centrifugation at 4 °C for 5 min at $1000 \times g$, and resuspended in 500 μL TRI Reagent (Sigma). RNA was extracted by phenol-chloroform precipitation. DNAse treatment was carried out with a DNA-free DNA Removal Kit (Invitrogen) according to the manufacturer's instructions. Library preparation was performed with the Illumina TruSeq RNA Kit. RNA sequencing was performed on the Illumina NextSeq500 by the Functional Genomics Centre Zurich (FGCZ).

**Computational RNA sequencing analysis**. Reads were quality-checked with FastQC[54]. Reads alignment to the reference genome "Mus_musculus.GRCm38.95" and read count was performed on the Support Users for SHell script Integration (SUSHI) framework[55], with the RSEMApp application. Pairwise comparisons were performed with the SUSHI application EdgeRApp (based on edgeR[56]). Principal component analysis and filtering of differentially expressed genes (DEGS, logFC > |2|, $p < 0.01$) were performed on R (version 3.6.1). The package pheatmap[57] was used to generate the heatmap of normalized FPKM. We performed GSEA[19] on significantly differentially expressed genes (logFC > |2|, $p < 0.01$) using the Bioconductor package fgsea with default parameters[58]. Genes were ranked based on p-value, and taking into account directionality of the fold change with the formula $ranking = -log10(P)/sign(log2ratio)$ (obtained from the blogpost http://genomespot.blogspot.co.at/2016/04/how-to-generate-rank-file-from-gene.html). The Hallmarks gene set collection from the Molecular Signatures Database[59] was imported in R with the package msigdbr. Cell type enrichment analysis of DEGs across conditions was performed on the web-based tool EnrichR[60,61].

**Droplet-based single-cell mRNA sequencing**. Control ($n = 3$) and D164A ($n = 3$) mice have sacrificed 0, 2 or 4d post injection, crypts were isolated as described[50]. Crypt fractions 3 and 4 were pooled and dissociated to single cells by incubating them with 5 mL pre-warmed TripLE Express Enzyme (ThermoFischer) in a 37 °C water bath for 5 min with frequent agitation. TripLE was inactivated with 50% FBS in Advanced DMEM/F12 (ThermoFischer). Single cells were pelleted at $180 \times g$ for 3 min and resuspended in 2 mL cold Intesticult (Stemcell Technologies Inc). Clumps were dissolved by pipetting up and down with a 1 mL pipet before filtering twice through a 40 μm filter. Cell viability and number were quantified in automatically with a Countess (Thermo Scientific). DropSeq workflow was performed as described[62] on a Nadia Instrument (Dolomite Bio). Sequencing was performed on the Illumina NovaSeq6000 with an SP Reagent Kit.

**Computational analysis of scRNaseq data**

*Dimensionality reduction and clustering*. Reads were quality-checked with FastQC. Raw data processing of fastq files was performed with the zUMIs pipeline[63] using the reference genome "Mus_musculus.GRCm38.95". scRNaseq analysis was performed with Seurat 3 on R 3.6.1[64]. Cells were filtered based on mitochondrial gene content, unique molecular identifier (UMI) counts were log-normalized according

to default Seurat settings. Scaling was performed on the variable genes (FindVariableFeatures, parameters: x.low.cutoff = 0.0125, x.high.cutoff = 3, y.cutoff = 0.5) regressing out UMI number and mitochondrial gene content. Samples were merged and joint analysis was performed with the package conos[65]. Cells were clustered with the Leiden community method (resolution = 1.3). The joint graph (in PCA space) was embedded in UMAP space and converted to a Seurat object. The stem cell and early progenitor clusters were subsetted and the log-normalized UMI counts were exported and used for the normalization algorithm and downstream analysis.

*Normalization algorithm*. Analysis of the control timecourse data revealed that the acute loss of one β-catenin allele caused a slight and transient downregulation of Wnt-targets, proliferation, and stem markers, which returned to normal expression levels by 4d pi. Thus, we devised a normalization of the mutant timecourse data to disentangle these effects from those induced by impaired N-terminal interactions, which were the focus of our investigation. For every timepoint (0, 2, and 4d pi), we divided the D164A single-cell expression (from data slot) of each gene by the mean expression of the corresponding gene in the control with the formula: ln[exp(*D164A*)/(*mean*(exp(*control*)]. The resulting three normalized D164A sparse matrices (one per timepoint) were used to create Seurat objects, which were merged, scaled, and visualized in UMAP space using dimensions 1:10.

*Diffusion maps and pseudotime analysis*. For diffusion maps and supervised pseudotime analysis, the normalized Seurat object was converted in a SingleCellExperiment. Diffusion maps were generated with the Bioconductor package destiny[66] using default parameters. Scores in the diffusion component 1 were compared using the Wilcoxon test (alternative: two sided) in R base. Supervised pseudotime analysis was performed with the Bioconductor package psupertime[29] on all genes or on mouse transcription factors only, setting scale to FALSE.

*Cell cycle scoring and classification*. Cell cycle scoring was performed with the CellCycleScoring algorithm from Seurat, using cell cycle-related genes from[28]. Wilcoxon test (alternative: two.sided) was used to compare G2M scores. To assign a measure of similarity to IESCs or TAs transcriptomic profiles, we trained a logistic regression (modification of the multinomial regression of MatchSCore2[67]) using IESC and TA markers extracted from a publicly available single-cell dataset[32]. Upon collecting the estimated probabilities of TA class memberships for each timepoint separately, we compared these probability distributions and tested the significance of the difference using the Kolmogorov-Smirnov test (alternative: two.sided).

*Gene set enrichment analysis*. We used the Seurat FindMarkers function (default parameters min.pct 0.25 and logfc.threshold 0.25) to compute differentially expressed genes between timepoints in our normalized D164A stem cell and early progenitor dataset. We then ranked genes with the formula $ranking = -log10(p\_val\_adj)/sign(avg\_logFC)$ and performed GSEA[19,59] on R using the Bioconductor package fgsea with default parameters[58]. The following gene set collections were imported in R with the package msigdbr: Hallmarks[59], Chemical and Genetic Perturbations (various contributions), KEGG[68,69], and gene ontology biological process[70]. Plots were generated with the R package ggplot2[71].

**ATAC-Seq**. Control ($n = 3$) and D164A ($n = 3$) mice were sacrificed 2d pi and processed independently for ATACSeq library preparation, following the protocol by[72]. Briefly, single-cell suspension of duodenal crypts was performed as for scRNAseq, which ensures virtually absent contamination from non-epithelial cells. 50,000 cells were lysed in 50 μl cold Lysis buffer (10 mM Tris-HCl, pH 7.4, 10 mM NaCl, 3 mM MgCl2, 0.1% IGEPAL CA-630). After centrifugation at $500 \times g$ for 10 min at 4 °C, cells were resuspended in 50 mL 1X Illumina transposition mix (25ul Tagment DNA Buffer, 2.5ul Tagment DNA Enzyme, 22.5 μl nuclease-free H2O) and incubated for 30 min at 37 °C on a shaker. Immediately following the transposition reaction, DNA was purified using the QIAgen MinElute PCR Purification Kit according to the manufacturer's instructions. The library was amplified with Nextera Sequencing primers in NEBNext Hot Start High-Fidelity 2X PCR master mix for 5 cycles. The appropriate number of additional PCR cycles was determined by qPCR. Amplified libraries were purified using the QIAgen MinElute PCR Purification Kit. DNA was eluted in 20 ml EB buffer and quantified and visualized for quality control with a 2200 TapeStation System (Agilent). Libraries were sequenced on an Illumina HiSeq2500 with paired-end 70 bp read configuration.

**Computational ATAC-Seq analysis**. Reads were quality-checked with FastQC. Adapters trimming with cutadapt, alignment on GRCh38.95 using Bowtie2 with default mapping parameters[73], as well as duplicate filtering with Picard and peak calling with MACS2 ($p < 0.01$) were performed on the ENCODE pipeline[74] supported by the SUSHI application AtacENCODEApp, for every biological replicate separately. The peak files were merged in BEDtools v2.29.2[75] and converted into a saf annotation, to which raw reads were mapped with the FeatureCounts function in the package rsubread[76]. The resulting count matrix was filtered for peaks with low coverage (minimum read count of 10 for each sample). The remaining counts were then normalized by total library size and TMM-derived normalization factors

calculated with edgeR[56]. Differential peaks between sample groups were identified using the exact test function in edgeR. Only peaks with a log fold change >1 and an $p$ value < 0.01 were considered as differentially accessible for further analysis. HOMER[39] was used for peak annotation and motif enrichment analysis. Motifs with Benjamini-corrected $q$-value < 0.001 were considered significantly enriched. bigWig files of control ($n = 3$) and D164A ($n = 3$) were combined into a single track for visualization in Integrative Genomics Viewer (IGV)[77]. Genes associated with a differentially accessible peak were ranked with the formula $ranking = -log10(Pvalue)/sign(log2FC)$ and used for GSEA with the R using the Bioconductor package fgsea with default parameters[58]. The significance of overlap between gene sets overrepresented in scRNAseq and in ATACSeq was quantified with hypergeometric test. Plots were generated with the R package ggplot2[71].

**Statistical analysis**. Statistical analysis and visualization were performed using R (Version 3.6.0, R Foundation for Statistical Computing Vienna, Austria), R Studio and Prism 8.2.0. Statistical significance tests were performed as described in each figure legend. Unless stated otherwise all tests were significant with $p$ value < 0.05. $p$ values > 0.05 were defined as not significant and are not presented in the graphs. The $p$ values provided were adjusted for multiple testing hypothesis according to the particular statistical method mentioned in the figure legend and in the corresponding part describing the computational analysis.

**Reporting summary**. Further information on research design is available in the Nature Research Reporting Summary linked to this article.

## Data availability

Data generated in this study have been deposited in the GEO database under the accession codes: GSE148941, GSE148942, GSE148940. The remaining data are available within the Article, Supplementary Information, or available from the authors upon request. Source data are provided with this paper.

## Code availability

The code used in this study is available at the public repository https://github.com/cocoborrelli/betacat (https://doi.org/10.5281/zenodo.4461476).

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

## Acknowledgements
S. Robine kindly shared the *villin-CreER^T2* strain with us. H. Clevers shared the Wnt3a L-cells. We are grateful to C. Cantù for valuable comments, and to S. Janjuha, W. Macnair, and B. Amati for advice. We appreciate discussions with members of the Basler and Moor lab, in particular with N. Doumpas and J. Hilchenbach. For technical help we thank E. Escher and the Functional Genomics Center Zurich. This work was supported by the Swiss National Science Foundation, the Swiss Cancer League, the University of Zurich Research Priority Program (URPP) "Translational Cancer Research" and the Kanton of Zürich. T.V. is supported by Czech Science Foundation grant 18-21466 S and is a fellow of the URPP Translational Cancer Research.

## Author contributions
C.B. designed and performed experiments, collected, analyzed and interpreted data and wrote the manuscript. T.V. initiated and conceived the research, designed and performed experiments, interpreted data, and wrote the manuscript. K.H., K.V., and G.M. assisted with the experiments. K.H., A.L., and L.V.R. helped with the bioinformatic analysis. A.G. and I.C.A. performed leukocyte profiling and intestinal permeability assay. G.H. and A.E.M. discussed the data and assisted with manuscript preparation. A.E.M. supported the research. K.B. initiated and supported the research, discussed the data, and assisted with manuscript preparation.

## Competing interests
The authors declare no competing interests.
