## [Peer Review File · Nature Communications]

Reviewers' Comments:

Reviewer #1:

Remarks to the Author:

This study from Konrad Basler's lab boils down to a functional genetic analysis of β -Catenin N-terminal and C-terminal transcription regulatory domains in murine intestinal epithelial cells. As major findings the authors confirm the vital importance of β -Catenin transcriptional activity in the intestinal epithelium, and report that C-terminally truncated β -Catenin and D164A mutant β -Catenin have vastly different consequences for intestinal epithelial tissue homeostasis and stem cell behavior. The finding of phenotypic differences between N-terminal and C-terminally mutated β -Catenin is neither surprising nor entirely novel since it was reported by the same lab before (Valenta et al, 2012, *Genes Dev* 25, 2631-2643). In contrast, the observed cellular and transcriptional changes ensuing from the D164A mutation indeed are highly interesting, and rightly a large part of the study is devoted to the characterization of the D164A mutant β -Catenin phenotype. By scRNA-seq, ATAC-seq, and bioinformatic analyses the authors substantiate the erroneous differentiation and stem cell depletion which had been observed histologically, and further document transient hyperactivation of proliferation. Moreover, changes in the unfolded protein response as well as elevated and ectopic JNK activity in D164A mutant β -Catenin animals become apparent. The experiments are well performed, the results are clearly described and for the most part of very high quality. Yet, this cannot belie the fact that virtually all the further conclusions and claims made by the authors have only correlative foundations and lack rigorous experimental testing and validation. Thus, the authors fail to clearly establish the primary defect caused by D164A mutant β -Catenin, to distinguish between what is cause and what is consequence, and to demonstrate in which regard activities associated with the N-terminus of β -Catenin are specifically and selectively involved in control of intestinal stem cell function.

Specific comments:

- I am having problems with expressions like "selectively governs" stem cell proliferation and identity, "fine-tuning β -catenin's transcriptional output", and "discrete regulation", as their meaning and the experimental basis for these statements remain obscure. To claim selective impact on stem cell features necessitates some kind of comparison and the demonstration that the same mutations had different consequences in another cell type. Such a comparative analyses was not undertaken in the current study. Fine tuning by D164A β -Catenin as opposed to the all-or-nothing effects of ΔC β -Catenin implies quantitative modulation of the same target genes which is not at all evident from the data shown. Looking at Figure 1C there is very little overlap among DEGs affected by the ΔC and D164A genes. The two mutants appear to affect entirely different gene expression programs. Where is the evidence that D164A β -Catenin increases or decreases Wnt target gene expression above or below, respectively, a critical level to warrant talking about fine tuning and "just right" signaling levels? Finally, what is the authors' concept of "discrete" regulation? That N-terminal and C-terminal interactors differentially contribute to the regulation of a given Wnt target gene? Or do they mean that N-terminal and C-terminal interactors selectively control distinct subsets for example of ubiquitous and intestinal stem cell-specific Wnt targets? Clarification of the terms mentioned above are needed and experimental back-up must be provided.

- A major shortcoming of the study is that the authors cannot distinguish between cell-autonomous and non-autonomous effects of D164A β -Catenin. Inflammation and immune cell infiltration could disrupt intestinal epithelial homeostasis as a secondary consequence to a primary defect caused by D164A β -Catenin. However it does not become evident what this defect could be and it is not unequivocally identified by the results of the study. One way to study cell-autonomous effects of D164A β -Catenin could be to ablate the floxed wildtype allele only upon establishing organoid cultures in the absence of an inflammatory or otherwise confounding cellular environment, and without a potentially priming effect by application of tamoxifen in vivo. To achieve this the authors could complement their organoid media with Wnt growth factors or the CV cocktail to boost Wnt signaling during the organoid initiation phase (Yin et al., *Nat Methods*. 2014 Jan; 11(1): 106-112).

- Can the authors definitively exclude the possibility that D164A β -Catenin impairs epithelial integrity? A breach of barrier function could lead to the observed immune cell infiltration and in consequence explain all the observed cellular and molecular changes seen in D164A β -Catenin animals.
- Dose effects of β -Catenin are known as evidenced by this study but also earlier work (Rudloff and Kemler, 2012, Development 139:3711-21). Especially its transcriptional activity in Wnt signaling seems to require sufficiently high protein levels. Therefore I would like to see Western blot experiments with intestinal epithelial cells and organoids to document equal expression of wildtype and mutant β -Catenin to exclude the possibility that protein level rather than the mutations account for the observed phenotypes.
- Based on previous work the authors insinuate that the phenotypic consequences of the D164A mutation are due to impaired interaction with Bcl9/Bcl9l. If this was the only molecular defect of the D164A mutant, why is it that Bcl9/Bcl9l double deficient mice do not phenocopy the D164A mutation and present with no or a much milder phenotype? The authors' attempt to explain this discrepancy in the Discussion is rather feeble and there is no reason why the argument of cellular plasticity should not similarly apply to β -Catenin mutants.
- From the pictures shown in Figure 2B coexpression of Lysozyme and Mucins cannot be deduced, and it remains unclear whether the D164A mutation leads to the appearance of "intermediate cells" or simply a massive differentiation of cells into nonetheless distinct branches of the secretory cell lineage. The panels should be replaced by higher quality images and examples of double positive cells should be highlighted. Besides, shouldn't the results of the scRNAseq support (or disprove) the existence of intermediate cells? Do Lyz/Muc2 double positive cells occur in the 4 day pi cell cluster and what are the proportions of double versus single positive cells compared to wildtype controls?
- The authors claim that JNK signaling mediates chromatin remodeling. This statement is based entirely on correlative observations. No causality was established. Functional validation is required by interfering with JNK signaling and demonstrating that this affects chromatin remodeling in D164A animals.

Minor issues:

- In the legend to Figure 4, subpanels 4D-F are incorrectly labeled.
- It should be explained what the boxed areas in Figures 1G-H; 5F and Extended Data Figure 3G specify.
- To allow for proper comparison and data evaluation magnified areas of both crypt and villus stainings for control as well as D164 animals should be shown in Extended Data Figure 3G, not just villus staining in controls and crypt staining in D164A animals.
- Please add numbers for features distinct and common to scRNAseq and ATACseq to the Venn diagrams shown in the left part of Extended Data Figure 4B.
- In their Methods section for the ATAC-seq the authors may want to check the units for the volumes used. They probably used microliters instead of milliliters for the experiment.

Reviewer #2:

Remarks to the Author:

The study reported in the manuscript entitled "Discrete regulation of β -catenin-mediated transcription governs identity of intestinal epithelial stem cells" by Borrelli et al., explore the role of site-specific transcriptional co-activators of β -catenin in maintaining stem cell renewal or contraction. The authors utilized mutant β -catenin alleles that impaired the requirement of N- or C-terminal co-effectors and by that govern the fate of the intestinal stem cell (ISC) by distinct activation of the Wnt signaling. The authors reported that while C-terminal co-activators are essential to induce Wnt target genes and resembled the knockout phenotype of β -catenin, the N-terminal co-effectors regulating cell proliferation and renewal of ISCs. Further examination of the N-terminal mutant (D164A) showed an excessive proliferation of ISC mediated by Myc-E2F activation. In addition, the mutated ISC activated the JNK pathway, which led to massive

differentiation and stem cell exhaustion, perhaps by immune cell involvement in the differentiation of ISCs.

While this study attempts to introduce a new role of N-terminal b-catenin activation in fine-tuning ISC fate by repressing JNK activation and Myc/E2F induced proliferation, further examination of the suggested mechanisms is needed. The single-cell RNA-Seq analysis in the study should be unbiased and include more QCs along with the analysis, including cell-type markers, cell subsets and states. In addition, as mentioned in the manuscript, previous single-cell genomics analyses of the small intestine were conducted and therefore a more concentrated work on the ISC compartment by utilizing stem cell-specific drivers should be included, such as the usage of the Lgr5-GFP-Cre-ERT2 model to assess the stem cell state of the mutated N-terminal b-catenin mice. Lastly, the involvement of immune cells in stem cell exhaustion was not well addressed and further follow-up experiments are required in order to understand if both the Myc-E2F-induced proliferation and JNK activation are related to immune-ISC crosstalk driven by N-terminal b-catenin mutated allele.

Overall, this study identifies a new role for N-terminal b-catenin co-factors in suppressing hyperproliferation and maintaining stem cell renewal in the gut. Still, the mechanism for its action is not clear. A model of activated JNK pathway could help to pinpoint the implications of Wnt signaling suppression of the JNK pathway in homeostasis, which is opposed to the known literature conducted in gut tumorigenesis (Sancho et al., 2009).

Major comments:

1. Single-cell analysis of the mutated b-catenin was conducted separately from the control mice. The authors compared the mutated to control mice in order to perform the analysis. An unsupervised analysis would help to determine ISC/progenitor states under the mutated b-catenin induction. To do so, the authors can combine all datasets and then look at DE genes of mutated b-catenin cells.
2. The immune involvement in the N-terminal mutated b-catenin is exciting and should be further investigated. Could the authors speculate on how b-catenin may be involved in immune recruitment? A recent study by Biton et al., 2018 showed the involvement of T helper cytokines in ISC renewal and differentiation fate, can the authors perform a further examination of the immune infiltrate shown in Fig. 4f? Does the mutated ISCs express high levels of Class II molecules?
3. In these lines, could the authors utilize immune-deficient mice to reduce gut inflammation and stem cell loss?
4. The authors described the phenotype in the small intestine, does the stem cell exhaustion occur in the colon as well? If so, does JNK signaling involved as well?

Minor comments:

5. Extended data Fig. 1c- could the author provide images from the colon? Do the inflammation is restricted to the small intestine or also to the colon
6. Fig. 1h – could the author provide Myc antibody stain instead of smFISH.
7. Organoids images are not clear and higher magnification could help to visualize the organoids state under different conditions. In addition, the quantification of organoids could help.
8. Extended data Fig. 3- Number of mice for each time point and genotype is missing (stated in the method section only).
9. Extended data Fig. 3a – The single-cell analysis of the small intestine includes known epithelial cell types but missing subsets of cells such as enteroendocrine and stem cell states (proliferative vs. non-proliferative). In addition, a UMAP divided by the mutated vs. control is missing. Could the authors provide the UMAP presentation of the different genotypes and time points?
10. Fig. 4F- could the author provide T cells (cd3, cd4 and cd8) stains? That could go well with the activation of immune cells and the pro-inflammatory signals shown in the manuscript.
11. Related to 4F- could the author provide stains for MHCII expression in the epithelium?
12. Fig. 3b – why the number of organoids in control at day4 is lower than day2?
13. Fig. 3c – the quantification of organoid numbers will help the reader.
14. Extended data Fig. 3b- could the authors provide 5 top markers of each cluster rather than showing known gene markers of each.

Reviewer #3:

Remarks to the Author:

The authors employ conditional mouse models to dissect the different contributions of N-terminal and C-terminal β -catenin binding partners to stem cell-driven homeostasis in the intestinal crypts. Selective loss of C-terminal interactions abolishes Wnt signaling, causing rapid loss of viable crypts. In contrast, selective loss of N-terminal interactions initially leads to hyperproliferation of Intestinal stem cells, followed by loss of stem cells through onset of differentiation programs likely driven by activation of JNK signaling.

In general, the findings of this study are interesting, well-founded and broadens our mechanistic insights into Wnt-regulated stem cell maintenance and differentiation in the intestine.

Critique

- 1) Maybe I missed this, but what are the possible (indirect) effects of the mutant alleles on cell adhesion? Could this be contributing to some of the phenotypes observed?
- 2) Ext fig1E – What are the total and nuclear β -catenin levels in the induced N/C mutants relative to controls?
- 3) Is there any possibility that the mutant alleles can be indirectly influencing Tcf/ β -catenin interactions due to altered folding?
- 4) Fig 1 – Use of Lgr5 reporter mice to better characterize direct effects on the intestinal stem cells and their progeny would have benefitted the study. There is currently an over-reliance on in situ hybridization markers to generate conclusions regarding stem cell/progenitor populations.
- 5) Fig 1E – The levels of Axin2 and Olfm4 expression visualized on the control tissues appears markedly lower than expected. I would suggest this may be due to compromised tissue and needs repeating to obtain a fair comparison with the mutant crypts. As above, the use of Lgr5 reporter mice would have greatly simplified the stem cell analyses here.
- 6) As I understand it, the study is restricted to evaluating the proximal SI only – are there region-specific differences observed?
- 7) What happens to Paneth cells @ early time-points following induction of the 164D allele? Given the changes in EphB expression observed, it would not be surprising if the Paneth cells quickly become mislocalised along the crypt-villus axis, potentially contributing to the observed stem cell phenotype.
- 8) Fig 1F – please confirm via q-PCR (or better via quantifying stem cell numbers using Lgr5 reporter mice).
- 9) Fig 1H – Plane of the control crypt is non-optimal – please repeat. Would it be worthwhile to more accurately quantify proliferation via EDU-injection/FACS (better yet, on sorted Lgr5+ stem cells and their progeny)?
- 10) Fig 2 – Given the interplay between Wnt and Notch in determining stemness vs. differentiation and the fact that the enhanced secretory cell phenotype is reminiscent of blockade of Notch signaling using inhibitors, is there any indirect effect of the mutant alleles on Notch signaling (perhaps via the JNK signaling route)?
- 11) Fig 3 – The fact that the mutant organoids cannot be passaged implies that there is indeed a rapid loss of stem cells. What happens to the Paneth cells in these mutant organoids? Is the differentiation phenotype recapitulated in the organoids?
- 12) Fig 3- Does Day 0 refer to organoids induced and then immediately harvested, or to non-induced organoids? If the latter, it is surprising that there is so little budding evident in the controls. If the former, I would then like to see non-induced organoids plated to generate an accurate comparison of the difference in organoid formation/budding extent. Again, the ability to selectively isolate Lgr5+ stem cells from control/mutant reporter mice would have facilitated a more direct evaluation of functional effects on the stem cells via organoid assay (and also a more accurate analysis of direct transcriptional changes within the stem cell compartment via ATAC-SEQ)

etc).

13) Fig 3 – Given that the ability of isolated crypts to survive as organoids ex vivo is dictated by their ability to rapidly circularize after plating, is it possible that the mutants simply decrease the efficacy of this initial process due to reduced adhesion? If the organoids are first generated before inducing mutant expression, perhaps the phenotype would be less severe/different?

14) Fig 4D – why is Lgr5 not included here?

15) Fig 5 – the link with JNK signaling is interesting- can this be validated by modulating this signaling pathway in intestinal organoids?

Reviewer #4:

Remarks to the Author:

In this manuscript the authors investigated how Wnt/beta-cat signaling defines the transcriptional outputs onto stem cell/epithelial progenitors to regulate intestinal homeostasis based on co-factor usage. They utilized elegant mice models to discern the fine tuning of Wnt signaling mediated by C or N-terminally co activators of beta-cat that are able to govern Wnt output solely from mutated allele. Additionally, other reference mouse models were used such as dm mutant mice combining both N and C coactivator deletion and beta-cat KO. They conclude that C-terminally recruited activators act as on-off-switch for the beta-cat mediated transcription of Wnt target genes, while N-terminal cofactors fine tune wnt signaling in order to ensure proper proliferation and self-renewal of intestinal stem cells. Overall this work is timely and will be of interest to the Nat Com readership, however across the manuscript the data has been selectively presented, controls not properly annotated, statistics provisionally added. These shortcomings, in my opinion, raise major concern(s) and need to be critically addressed. Therefore my review, at this point, focuses mainly on concerns regarding consistency of data presentation (taking into an account the information authors provided).

Major concerns:

1. In figure 1 authors conclude that expression profile of N- vs C-terminal beta-cat co-factors is distinct and expression profile of delta C is similar to KO based on analyses of bulk rna sequencing of multidimensional data set of several groups n=2 deltaC, n=3 D164, n=2 Ko, n=3 dm, n=3 control. This is a rather small n per group and yet authors choose to present only one sample of each group in Figure1B. The authors should better present all samples used for sequencing individually to reflect biological variance among the animals belonging to each group. On the following panel 1C the cross comparison of this multidimensional data set is presented. Can authors explain and justify (also in the manuscript) why dm mutant mice that are missing both N and C terminally co-activators have only around 50 DEGs. One would expect broader changes in these animals compared to delta C or D164A mutant. Additionally, the reference 17 does not refer to dm mutant as annotated in manuscript. Please present the comparisons between groups as wenn diagrams for easier visualization, this is very well achievable among 4 groups. The biological variance between the samples belonging to each group is obvious across the sample of genes presented in 1D heatmap. Clearly not all the genes change in the same direction within the same group (see mm7, sox9, tnfs19...). This leaves me wonder whether 2-3 animals per group are representative sample size that authors repeatedly use, even in downstream analyses such as in situ hybridisation. Secondly quantification of Rna dots by this particular technique is rather challenging if not arbitrary even with the software obtained from company. Thirdly, the statistics can not be performed on 2 biological replicates, therefore either include more animals or exclude the statistics. As adding more biological replicates to sequencing data might be time consuming and expensive the authors should critically perform qRT pcr on at least 3-5 animals per group (including all the groups: dm, wt, deltaC, D164A, KO) for at least Lgr5, Olfm4, Ascl2, axin2 and cd44 and IHC for c-myc (and present those on the same graph).

2. Delta C mutant in supplementary figure 1E appears to be very different than control at 2dpi. Multiple crypts are undergoing crypt fission, a sign of increased stem cell activity which contrast with conclusions of figure 1 showing loss of stem cells and proliferation at 2dpi. This inconsistency together with such limited number of animals used for assays aligns with previous concern

questioning consistency of the delta C phenotype or reflecting inconstancy or recombination? Therefore, in my opinion, phenotypes of delta C and D164A mutants would need to be further investigated to justify the author's conclusions.

3. In figure 3 the authors focus on D164A mutant and observe on day 4pi loss of stem cells accompanied with secretory hyperplasia judged by Lyz/Pas staining and conclude that aberrant stem cell activation at day 2 leads to aberrant secretory cell differentiation into intermediate double positive Lyz/Pas cells (Figure 2B). However in supplementary figure 4A this cell type is not annotated nor its presence acknowledged in manually selecting cell clusters. This seems odd as authors in material and methods state to use crypt fraction for single cell sequencing and from figure 2 observe the majority of the double positive Lys/pas cells in the crypt whose presence is even acknowledged in the main text of the manuscript. Please clarify to me how is it possible to observe an increase of manually adjusted Paneth and goblet cell clusters (composed of fully differentiated Paneth or Goblet cells respectively with no overlap in expression Supplementary Fig3B) within the crypts of D146A while the crypts are composed of double positive secretory progenitors (Fig 2B) that were not accounted for?

4. From the text it is not obvious from the first read what animals are used as a control for the single cell sequencing experiment presented on figure 4 and supplementary figure 3. Please include the genotype and explain in main text/figure legend.

5. On figure 4A is obvious that only a selection of single cells is presented (SCEP account 25-50% of the all cell analyzed, Supplementary figure 3 C). Please clarify whether the z score in figure 4C was calculated from the whole data set or preselected cells and if later the case specify the criteria in detail.

6. Please provide better representation of Creb3l3 rna in situ and add quantification of crypts expressing it. In supplementary figure only 1 crypt (from 2 shown) is expressing the transcript, while the other doesn't and the expressing crypt does not belong to the crypt villus unit therefore it is questionable at which plane of the crypt villus unit this crypt belongs to.

7. What control (genotype) was used for generating the data presented on figure 5?

We would like to thank all reviewers for their questions and valuable comments. We are very pleased about their positive view concerning our manuscript. Following the reviewer's recommendations, we have now comprehensively addressed all points raised and include additional experiments, in order to enhance the impact of our work. All comments are addressed in detail in the point-by-point reply below.

We dedicated great effort to show that the structural function of β -catenin mutants, and thus the integrity of the intestinal barrier, are not altered (new Figures: Fig.S1G, Fig.5G). An equally important effort was directed towards the dissection of the immune infiltration in the intestine of β -catenin-D164A mutants (newly added Fig.5C-F, Fig.S6A,B). We also increased the number of biological replicates, independently validated our main findings with qRT-PCR (for example Fig.S2C and Fig 5B), and added more *in vitro* experiments on organoids (Fig.3, Fig.S4). Additionally, we analysed the secretory hyperplasia/mis-differentiation observed in D164A mutants in more detail (replaced panels in Fig.2B, new analysis shown in Fig.S5E,F). Last but not least, we added new data showing phenotypic similarities between loss of Bcl9/9l (under Ctnnb1 hemizygous conditions) and our β -catenin-D164A mutant (Fig.7A,B).

As mentioned, we substantially extended the number of figures. The main changes are:

- Fig.2B, Fig.3C,D were improved with new panels (new stainings)
- Fig.5 contains entirely new data (panels B-G), Fig.5A was formerly Fig.S3E
- Fig.6 is former Fig.5
- Newly added Fig.7A,B
- New panels/data in Supplementary Figures (Fig.S1G,F and Fig.S2A-F and Fig.S3A-D and Fig.S4C-E and Fig.S5A,C-F and Fig.S6A-B)
- Other panels/figures were re-numbered/re-arranged if required: for instance Fig.S7 is former Fig.S4, Fig.S4A,B are former panels of Fig.S2)

The text changes throughout the manuscript added to improve the clarity of the portrayed messages or reflecting new data are highlighted in blue.

We believe that all these changes and additions addressed the reviewers' concerns and together resulted in a strongly improved dataset and manuscript.

Note : Reviewer's comments are in ***Bold Italics***

Reviewer #1 (Remarks to the Author):

This study from Konrad Basler's lab boils down to a functional genetic analysis of β -Catenin N-terminal and C-terminal transcription regulatory domains in murine intestinal epithelial cells. As major findings the authors confirm the vital importance of β -Catenin transcriptional activity in the intestinal epithelium, and report that C-terminally truncated β -Catenin and D164A mutant β -Catenin have vastly different consequences for intestinal epithelial tissue homeostasis and stem cell behavior. The finding of phenotypic differences between N-terminal and C-terminally mutated β -Catenin is neither surprising nor entirely novel since it was reported by the same lab before (Valenta et al, 2012, Genes Dev 25, 2631-2643). In contrast, the observed cellular and transcriptional changes ensuing from the D164A mutation indeed are highly interesting, and rightly a large part of the study is devoted to the characterization of the D164A mutant β -Catenin phenotype. By scRNA-seq, ATAC-seq, and bioinformatic analyses the authors substantiate the erroneous differentiation and stem cell depletion which had been observed histologically, and further document transient hyperactivation of proliferation. Moreover, changes in the unfolded protein response as well as elevated and ectopic JNK activity in D164A mutant β -Catenin animals become apparent. The experiments are well performed, the results are clearly described and for the most part of very high quality. Yet, this cannot belie the fact that virtually all the further conclusions and claims made by the authors have only correlative foundations and lack rigorous experimental testing and validation. Thus, the authors fail to clearly establish the primary defect caused by D164A mutant β -Catenin, to distinguish between what is cause and what is consequence, and to demonstrate in which regard activities associated with the N-terminus of β -Catenin are specifically and selectively involved in control of intestinal stem cell function.

Specific comments:

-I am having problems with expressions like "selectively governs" stem cell proliferation and identity, "fine-tuning β -catenin's transcriptional output", and "discrete regulation", as their meaning and the experimental basis for these statements remain obscure. To claim selective impact on stem cell features necessitates some kind of comparison and the demonstration that the same mutations had different consequences in another cell type. Such a comparative analyses was not undertaken in the current study.

We agree with this point and have, where appropriate, adjusted the text and/or provided additional data to support the claim. Regarding the stem cell specificity, for instance, we now document unaltered phenotype of Paneth and goblet cells 2d pi (Fig. S3C-D). Moreover, by comparative cell-cycle analysis, we show in Fig.S5F that the proliferation of Paneth cells and crypt enterocytes is unaltered, in contrast to that of stem cells and early progenitors (SCEP cells, Fig.4C, Fig 1E and newly added Fig S2F).

At the same time, we believe that the impact of β -catenin mutations is preferentially exerted on stem cells and early progenitors. In general, assessing the impact of altered transcriptional outputs of β -catenin on non-crypt cell types is not considered necessary, because Wnt signaling is only active in crypts – a view supported by work from many other labs. Furthermore, as it is shown in Fig.S1E,G, the β -catenin mutant proteins we use are present at adherens junctions and thus maintain their structural role, which is important for all epithelial cell types, consistent with what we previously showed (Valenta et al. 2011).

-Fine tuning by D164A β -Catenin as opposed to the all-or-nothing effects of ΔC β -Catenin implies quantitative modulation of the same target genes which is not at all evident from the data shown. Looking at Figure 1C there is very little overlap among DEGs affected by the ΔC and D164A genes. The two mutants appear to affect entirely different gene expression programs. Where is the evidence that D164A β -Catenin increases or decreases Wnt target gene expression above or below, respectively, a critical level to warrant talking about fine tuning and “just right” signaling levels? Finally, what is the authors’ concept of “discrete” regulation? That N-terminal and C-terminal interactors differentially contribute to the regulation of a given Wnt target gene? Or do they mean that N-terminal and C-terminal interactors selectively control distinct subsets for example of ubiquitous and intestinal stem cell-specific Wnt targets? Clarification of the terms mentioned above are needed and experimental back-up must be provided.

The reviewer is right; we adjusted the text. In addition to changes highlighted in blue within the text, we changed the wording “discrete regulation” into “differential regulation” or “selective modulation”, which reflect our results more accurately. Below we briefly summarize our ideas. We believe that C- vs. N-terminal interactors are involved in the regulation of core β -catenin targets, such as Wnt-target genes, proliferation and stem cells markers at 2d pi. These targets are influenced by both D164A and C-terminal mutations, but they respond very differently (Fig.1D, and newly added qRT-PCR validation on more biological replicates in Fig. S2C). Indeed, we show upregulation in D164A and downregulation in ΔC mutants by RNAseq (Fig.1D), *in situ* hybridization (Fig. 1E,G) and qRT-PCR (Fig.S2C), as well as protein stainings (newly added staining in Fig. S2F).

On the other hand, we believe that the regulation (direct or indirect) of a distinct gene subset might occur, inducing secretory hyperplasia and immune recruitment specifically in D164A mutants at 4d pi. The small overlap in DEGs between these two mutants is mostly due to immune genes coming from infiltrating immune cells in the D164A mutant (Fig.S2D), which are now extensively studied in the new figures – Fig.5C-F, Fig.S6A,B).

-A major shortcoming of the study is that the authors cannot distinguish between cell-autonomous and non-autonomous effects of D164A β -catenin. Inflammation and immune cell infiltration could disrupt intestinal epithelial homeostasis as a secondary consequence to a primary defect caused by D164A β -Catenin. However it does not become evident what this defect could be and it is not unequivocally identified by the results of the study. One way to study cell-autonomous effects of D164A β -Catenin could be to ablate the floxed wildtype allele only upon establishing organoid cultures in the absence of an inflammatory or otherwise confounding cellular environment, and without a potentially priming effect by application of tamoxifen *in vivo*. To achieve this the authors could complement their organoid media with Wnt growth factors or the CV cocktail to boost Wnt signaling during the organoid initiation phase (Yin et al., Nat Methods. 2014 Jan; 11(1): 106–112).

This is an interesting point. We followed the recommendation of the reviewer and tried to perform various organoid-based experiments (Fig.S4A-E). Despite a substantial effort we were not able to propagate organoids *in vitro*, if hemizygous for β -catenin, neither constitutively (Ctnnb1-KO/wt), nor when recombination was induced *in vivo* or *in vitro* (Ctnnb1-wt/flox) (Fig. S4A-D). Enhancing Wnt-signaling activity by addition of Wnt3a-conditioned medium (Fig.S4E), or GSK3 inhibitor CHIR99021 (not shown) did not rescue the lethality of hemizygous or mutant organoids.

Thus, we were not able to characterize our mutants *in vitro*, in the absence of a supporting niche. Of note, these results provided yet another line of evidence suggesting that, despite being a great experimental model, organoids have certain limitations and fail to completely recapitulate the situation *in vivo*. An example with respect to Wnt signaling and the requirement for cofactors is the following: *in vivo* deletion of Bcl9/Bcl9l in the epithelia does not impair intestinal homeostasis, however it is not possible to establish intestinal organoids from such intestinal crypts Bcl9/Bcl9l (as shown by Gay et al., 2019). We also examined the functionality of the intestinal barrier, analysed in more details the immune infiltration in D164A mutant and the extended focus to secretory cells. We believe these are steps that will enable future work to discriminate between cell-autonomous and non-autonomous effects.

-Can the authors definitively exclude the possibility that D164A β -Catenin impairs epithelial integrity? A breach of barrier function could lead to the observed immune cell infiltration and in consequence explain all the observed cellular and molecular changes seen in D164A β -Catenin animals.

In the revised version of our manuscript, we are addressing this very important comment by two types of data. The immunoblot in Fig.S1F shows that 2 days after induction of recombination of the conditional allele (*Ctnnb1-flox*), the wt protein encoded by this allele is completely eliminated in the Δ C mutant. The truncated protein (i.e. Δ C) corresponds to the band with increased mobility (12 kDa shift). Since we used the same conditional allele and experimental design for the D164A mutant, we also expect that at 2d pi, the only expressed form of β -catenin is the mutant one (in D164A the single amino acid change is not distinguishable by immunostaining or -blotting). Fig.S1G shows that the mutant β -catenin proteins are present at the cellular membranes and colocalize with the epithelial cell adhesion molecule Epcam, indicating normal composition and function of adherens junctions.

Additionally, we probed the functionality of the intestinal barrier (and its possible leakiness) by FITC-dextran administration. As shown in Fig.5G, epithelial barrier integrity is unaltered in control animals and in both C- and N-terminal mutants at 2d pi, and for control and D164A mutant also at 4d pi (ΔC animals suffer from crypt atrophy and reach humane endpoint 4d pi). Hence, we conclude that the observed phenotypes (2d pi and 4d pi) are not a consequence of compromised intestinal barrier function.

-Dose effects of β -Catenin are known as evidenced by this study but also earlier work (Rudloff and Kemler, 2012, Development 139:3711-21). Especially its transcriptional activity in Wnt signaling seems to require sufficiently high protein levels. Therefore, I would like to see Western blot experiments with intestinal epithelial cells and organoids to document equal expression of wildtype and mutant β -Catenin to exclude the possibility that protein level rather than the mutations account for the observed phenotypes.

We added a western blot showing cytoplasmic and nuclear levels of β -catenin in the intestinal crypt cells isolated 2d pi (Fig.S1F). The levels of wildtype and D164A β -Catenin are similar. As expected from the phenotype, we see less nuclear cytoplasmic β -catenin- ΔC .

In more detail, we used an antibody recognizing the N-terminus of β -catenin to probe the levels. In the ΔC mutant, no full length β -catenin was seen. Thus, at the time the samples were taken no residual wt protein was present. By extension, we assume that in the D164A mutant also only the mutant is present. In the case of the D164A mutant, both the nuclear and cytoplasmic levels were comparable to the control. The lower nuclear amount of ΔC β -catenin probably reflects the fact that most of the stem cells and early progenitors (i.e. cells actively responding to Wnt ligands and thus with nuclear β -catenin) are lost at 2d pi. Importantly, the cytoplasmic β -catenin- ΔC levels are the same as in the control, suggesting normal expression and processing of the mutant protein. Hence, we conclude that the proteins levels do not account for the observed phenotypes in the mutants.

-Based on previous work the authors insinuate that the phenotypic consequences of the D164A mutation are due to impaired interaction with Bcl9/Bcl9l. If this was the only molecular defect of the D164A mutant, why is it that Bcl9/Bcl9l double deficient mice do not phenocopy the D164A mutation and present with no or a much milder phenotype? The authors' attempt to explain this discrepancy in the Discussion is rather feeble and there is no reason why the argument of cellular plasticity should not similarly apply to β -Catenin mutants.

We address this very important point by new data in Fig.7A,B. As the reviewer mentioned, epithelial loss of Bcl9/Bcl9L does not affect the intestinal homeostasis. The Bcl9/Bcl9L-KO mice lack the Bcl9/9L- β -catenin transcriptional contribution/co-activation, but feature otherwise intact β -catenin expression. The D164A mutant lacks the Bcl9/9L- β -catenin interaction and additionally possesses only one (mutated) β -catenin allele (upon recombination of the conditional, i.e. flox allele). Hence, we set out to investigate whether the β -catenin hemizygous state explains the phenotypic difference between the two models. Indeed, when we mimicked the condition in our N-terminal (D164A) β -catenin mutant, by the combination of Bcl9/9L loss together with an acute reduction of β -catenin levels (*villin-CreER^{T2}; Bcl9^{flox/flox}; Bcl9^{flox/flox}; Ctnnb1^{wt/flox}*), we could recapitulate the D164A phenotype: lethal crypt atrophy, secretory hyperplasia and villus shortening. Thus, the role of the β -catenin-Bcl9 interaction becomes essential in situations where epithelial homeostasis and plasticity is challenged by perturbations of Wnt signaling. This was independently shown by other groups, for example, upon the Porcupine inhibition (Gay *et al.*, 2019). Such a phenomenon is mirrored in Wnt-addicted tumor situations, where high levels of Wnt signaling are needed, and every reduction (i.e. Bcl9/9L loss) has important phenotypic consequences (Moor *et al.*, 2015, Gay *et al.*, 2019, Mieszczanek *et al.*, 2019). In contrast to these two extreme situations, the wild type can maintain homeostasis upon loss of Bcl9/9L most likely via epithelial plasticity mechanisms. These aspects are further discussed in a new paragraph in the Discussion section.

The newly added Fig.7A,B

- A) Survival plot of *villin-CreER^{T2};Ctnnb1^{wt/flox}* (control, n=3), *villin-CreER^{T2};Ctnnb1^{D164A/flox}* (D164A n=5), *villin-CreER^{T2};Bcl9^{flox/flox};Bcl9L^{flox/flox};Ctnnb1^{wt/flox}* (n=3) and *villin-CreER^{T2};Bcl9^{flox/flox};Bcl9L^{flox/flox};Ctnnb1^{wt/wt}* animals (n=3). $p < 0.001$ as calculated by log-rank test.
- B) Histology sections of *villin-CreER^{T2};Bcl9^{flox/flox};Bcl9L^{flox/flox};Ctnnb1^{wt/flox}* and *villin-CreER^{T2};Bcl9^{flox/flox};Bcl9L^{flox/flox};Ctnnb1^{wt/wt}* animals. Concomitant deletion of *Bcl9*, *Bcl9L* and one copy of β -catenin induces crypt atrophy and villus shortening. Timepoint: 6d pi. Scale bars, 20 μ M.

-From the pictures shown in Figure 2B coexpression of Lysozyme and Mucins cannot be deduced, and it remains unclear whether the D164A mutation leads to the appearance of “intermediate cells” or simply a massive differentiation of cells into nonetheless distinct branches of the secretory cell lineage. The panels should be replaced by higher quality images and examples of double positive cells should be highlighted. Besides, shouldn’t the results of the scRNAseq support (or disprove) the existence of intermediate cells? Do Lyz/Muc2 double positive cells occur in the 4 day pi cell cluster and what are the proportions of double versus single positive cells compared to wildtype controls?

We replaced the panel with new ones (new stainings) – Fig.2B. Additionally we show unchanged secretory differentiation (Paneth vs. goblet cells) in ΔC and D164A mutants 2d pi (Fig.S3C,D). Consistent with the immunostaining, detailed analysis of scRNAseq data revealed the presence of secretory mis-differentiation (secretory cells simultaneously showing features of both Paneth and goblet cells) in D164A mutants at 4d pi, but not at 2d pi, where the cells are mostly scattered along the two axes (Fig.S5E)

-The authors claim that JNK signaling mediates chromatin remodeling. This statement is based entirely on correlative observations. No causality was established. Functional validation is required by interfering with JNK signaling and demonstrating that this affects chromatin remodeling in D164A animals.

We agree with the reviewer that our statements concerning JNK chromatin remodeling are not based on direct causal observation. We changed the text to reflect this. However, we stand by our interpretation. JNK was shown in various models to be able to induce chromatin remodeling (reviewed in Madrigal and Alasoo, 2018, Klein *et al.*, 2013). Moreover, new links between Wnt signaling and JNK signaling were recently established (Harmston *et al.*, 2020). Moreover, our ATACseq data suggests JNK-mediated activity is involved. As shown in Fig.6C and Fig.S7A (and

partially by other figures) the JNK/Jun/Fos-associated binding motifs (BATF, Junb, AP-1, Fos) are highly associated with altered ATAC-seq peaks. In addition, some JNK target genes (e.g. Muc13, Fig.6A) showed peaks associated with an increased expression. Although it could still be just a correlation, we think that together with other published data (Moor *et al.*, 2018) our observation points toward causality, albeit indirectly.

Minor issues:

-In the legend to Figure 4, subpanels 4D-F are incorrectly labeled.

We corrected it.

-It should be explained what the boxed areas in Figures 1G-H; 5F and Extended Data Figure 3G specify.

We added the information for all relevant panels in all figure legends.

-To allow for proper comparison and data evaluation magnified areas of both crypt and villus stainings for control as well as D164 animals should be shown in Extended Data Figure 3G, not just villus staining in controls and crypt staining in D164A animals.

We added this information. Insets showing both crypt and villus area are shown in Fig.5A (former Fig. S3G).

-Please add numbers for features distinct and common to scRNAseq and ATACseq to the Venn diagrams shown in the left part of Extended Data Figure 4B.

Numbers were added to the Venn diagram – now Fig.S7B (former Fig. S4B).

-In their Methods section for the ATAC-seq the authors may want to check the units for the volumes used. They probably used microliters instead of milliliters for the experiment.

We corrected this, thanks!!

Reviewer #2 (Remarks to the Author):

The study reported in the manuscript entitled “Discrete regulation of β -catenin-mediated transcription governs identity of intestinal epithelial stem cells” by Borrelli et al., explore the role of site-specific transcriptional co-activators of b-catenin in maintaining stem cell renewal or contraction. The authors utilized mutant b-catenin alleles that impaired the requirement of N- or C-terminal co-effectors and by that govern the fate of the intestinal stem cell (ISC) by distinct activation of the Wnt signaling. The authors reported that while C-terminal co-activators are essential to induce Wnt target genes and resembled the knockout phenotype of b-catenin, the N-terminal co-effectors regulating cell proliferation and renewal of ISCs. Further examination of the N-terminal mutant (D164A) showed an excessive proliferation of ISC mediated by Myc-E2F activation. In addition, the mutated ISC activated the JNK pathway, which led to massive differentiation and stem cell exhaustion, perhaps by immune cell involvement in the differentiation of ISCs. While this study attempts to introduce a new role of N-terminal b-catenin activation in fine-tuning ISC fate by repressing JNK activation and Myc/E2F induced proliferation, further examination of the suggested mechanisms is needed. The single-cell RNA-Seq analysis in the study should be unbiased and include more QCs along with the analysis, including cell-type markers, cell subsets and states. In addition, as mentioned in the manuscript, previous single-cell genomics analyses of the small intestine were conducted and therefore a more concentrated work on the ISC compartment by utilizing stem cell-specific drivers should be included, such as the usage of the Lgr5-GFP-Cre-ERT2 model to assess the stem cell state of the mutated N-terminal b-catenin mice. Lastly, the involvement of immune cells in stem cell exhaustion was not well addressed and further follow-up experiments are required in order to understand if both the Myc-E2F-induced proliferation and JNK activation are related to immune-ISC crosstalk driven by N-terminal b-catenin mutated allele. Overall, this study identifies a new role for N-terminal b-catenin co-factors in suppressing hyperproliferation and maintaining stem cell renewal in the gut. Still, the mechanism for its action is not clear. A model of activated JNK pathway could help to pinpoint the implications of Wnt signaling suppression of the JNK pathway in homeostasis, which is opposed to the known literature conducted in gut tumorigenesis (Sancho et al., 2009).

Major comments:

-Single-cell analysis of the mutated b-catenin was conducted separately from the control mice. The authors compared the mutated to control mice in order to perform the analysis. An unsupervised analysis would help to determine ISC/progenitor states under the mutated b-catenin induction. To do so, the authors can combine all datasets and then look at DE genes of mutated b-catenin cells.

As mentioned in the methods section, the single-cell RNAseq datasets we merged before clustering: “Samples were merged and joint analysis was performed with the package conos⁶⁵. Cells were clustered with the Leiden community method (resolution=1.3). The joint graph (in PCA space) was embedded in UMAP space and converted to a Seurat object. The stem cell and early progenitor clusters were subsetted and the log-normalized UMI counts were exported and used for the normalization algorithm and downstream analysis.” Therefore, unsupervised clustering and annotation reflect the similarity in cell states irrespective of genotype and timepoint. After an initial analysis in which we performed pairwise comparisons (between the same timepoint across genotypes, now reported in Fig.S5C,D, and same genotype across timepoints), we devised a custom normalization procedure to “filter out” the effect induced by the acute reduction of β -catenin and only focus on the transcriptomic changes induced by impaired N-terminal interactions. We re-formulated the text to clarify this part. Importantly, the differential gene expression analysis

between D164A and control at 2d and 4d pi now reported in Fig.S5C-D, are in the line and further validate the bulk RNAseq data (and the confirmatory real-time qRT-PCR).

-The immune involvement in the N-terminal mutated b-catenin is exciting and should be further investigated. Could the authors speculate on how b-catenin may be involved in immune recruitment? A recent study by Biton et al., 2018 showed the involvement of T helper cytokines in ISC renewal and differentiation fate, can the authors perform a further examination of the immune infiltrate shown in Fig. 4f? Does the mutated ISCs express high levels of Class II molecules?

We investigated this very interesting aspect in more detail not only by careful reanalysis of our data, but additionally by performing new experiments. We now show enhanced expression of genes involved in antigen presentation (namely MHC II class molecules) as indicated in Fig.5C in detail. Notably, we performed FACS-based experiments and cytokine profiling that revealed enhanced infiltration by Cd4 T-cells, monocytes and neutrophils specifically in D164A mutants 4d pi (Fig.5E,F and Fig.S6A).

We determined substantially, but not significantly, increased expression of IFN γ , TNF α and IL-17 (Fig.S6B) which reflect the increased IFN γ , TNF α -signalling in stem cells shown by GSEA in our scRNASeq data. The possible involvement of those cytokines in the observed phenotype, specifically loss of self-renewal and secretory differentiation, is discussed within the newly added text. Our data are strikingly in line with what was observed by Biton *et al.* 2018, and also recently by Sato *et al.*, 2020. The connection between Wnt/ β -catenin-signaling and immune infiltration is now discussed in the conclusion section of our manuscript and is definitely interesting for follow-up studies.

-In these lines, could the authors utilize immune-deficient mice to reduce gut inflammation and stem cell loss?

We thank for this interesting suggestion. However, it will be very time consuming and challenging to combine our experimental model with immune-deficient animals. Moreover, by eliminating the immune infiltration and later inflammation, we may obtain data which do not reflect the situation *in vivo*, as intricate as it may be, which is however our primary interest. As mentioned above, recent work showed the impact of IFN γ on intestinal stem cell homeostasis and differentiation (Sato et al., 2020). Since we determined that enhanced immune infiltration is not a consequence of compromised intestinal barrier, but rather induced by crypt-derived signals upon perturbation of N-terminal β -catenin interactions, we believe that it represents an important aspect of the phenotype.

-The authors described the phenotype in the small intestine, does the stem cell exhaustion occur in the colon as well? If so, does JNK signaling involved as well?

The onset of the phenotype is much faster in the small intestine than in the colon, as we did not observe any overt morphology nor altered proliferation in the colonic epithelium (newly added Fig.S3A,B). This phenomenon probably reflects higher renewal dynamics of small intestine and is in line with our previous observations indicating that the effects of blocking transcription-specific outputs of β -catenin are first apparent in the small intestinal epithelium (Valenta *et al.* 2016). The lethal loss of small intestinal function results in humane endpoint before substantial effects on the colon epithelium were visible. Hence, we primarily focused on the small intestine.

Minor comments:

-Extended data Fig. 1c- could the author provide images from the colon? Do the inflammation is restricted to the small intestine or also to the colon

As mentioned above, images of colon are now provided (Fig.S3A,B)

-Fig. 1h – could the author provide Myc antibody stain instead of smFISH.

We repeatedly tried to stain for cMyc by immunohistochemistry (using various antibodies), however without any success. We discussed this issue with Prof. Bruno Amati (Department of Experimental Oncology, IFOM-IEO, Milano, Italy), an expert in the cMyc field. He confirmed, that staining for cMyc in primary intestinal tissue is very tricky and almost never works. We believe that the combination of smFISH with strong Myc-E2F signatures observed in the transcriptomic data provide strong evidence concerning the changes in cMyc expression.

-Organoids images are not clear and higher magnification could help to visualize the organoids state under different conditions. In addition, the quantification of organoids could help.

This information is now added (Fig.3C, Fig.S4A-E).

-Extended data Fig. 3- Number of mice for each time point and genotype is missing (stated in the method section only).

This information is now added to figure legend (now Fig.S5).

-Extended data Fig. 3a – The single-cell analysis of the small intestine includes known epithelial cell types but missing subsets of cells such as enteroendocrine and stem cell states (proliferative vs. non-proliferative). In addition, a UMAP divided by the mutated vs. control is missing. Could the authors provide the UMAP presentation of the different genotypes and time points?

As mentioned in the methods section, we performed a merged analysis of all datasets. This allows for unsupervised clustering of the cells irrespective of genotype and time point, but rather based on similarity between cell states. We believe that this kind of analysis is more appropriate than split analysis (control vs. mutant) in a situation, like the one presented here, where cell types and cell states are rapidly changing. Nevertheless, we improved and extended our analysis, and we now provide control vs mutant differential gene expression at 2d and 4d pi, as well as comparative cell cycle analysis of Paneth cells and crypt enterocytes, and profiling of intermediate cells (Fig.S5C-F). Thereby, we confirm that the main impact of mutant β -catenin is in stem cells and early progenitors. This is not so surprising, since these cells have repeatedly been shown to exhibit active Wnt signalling and nuclear β -catenin (Valenta *et al.*, 2016, vanEs *et al.*, 2012, Fevr *et al.*, 2007).

-Fig. 4F- could the author provide T cells (cd3, cd4 and cd8) stains? That could go well with the activation of immune cells and the pro-inflammatory signals shown in the manuscript.

This information is now provided in the newly performed FACS-based experiments (Fig.5E,F and Fig.S6A,B).

-Related to 4F- could the author provide stains for MHCII expression in the epithelium?

The information concerning MHCII expression is now part of the Fig.5C,F. Unfortunately, the staining suffered of high background/low specificity.

- Fig. 3b – why the number of organoids in control at day4 is lower than day2?

The reason is that one copy of wt β -catenin (i.e. hemizygous status) is not sufficient to promote the growth of organoids *in vitro*. Despite a substantial effort we were not able to propagate organoids, if hemizygous for β -catenin, *in vitro*, neither constitutively (*Ctnnb1-KO/wt*), nor when recombination was induced *in vivo* (before crypt isolation) nor *in vitro* (*Ctnnb1-wt/flox*) (Fig. S4A-D). Even addition of external Wnt3a did not support the growth of hemizygous organoids (i.e. *villin-CreER^{T2};Ctnnb1^{wf/flox}* after 4OHT addition, featured in Fig. S4E). This observation makes the increased organoid formation rate in crypts isolated 2d pi from D164A ever so striking. However, it also meant that organoids cannot be used, in this genetic setting, to study the effect of β -catenin mutations. This is in striking contrast to situation *in vivo*, where constitutively hemizygous *Ctnnb1-KO/wt* animals are perfectly viable and fertile. Also, the conditional removal of one copy of β -catenin does not influence the intestinal homeostasis (*villin-CreER^{T2}; Ctnnb1-wt/flox* after tamoxifen administration). Hence, we used *villin-CreER^{T2};Ctnnb1-wt/flox* animals as an appropriate control for our *in vivo* studies.

The discrepancy between organoids and the *in vivo* situation strongly resembles the case of the epithelial elimination of Bcl9/Bcl9l: while animals lacking Bcl9/Bcl9l are perfectly viable, it is not possible to generate intestinal organoids from such animals, and organoids die after loss of Bcl9/Bcl9l *in vitro* (Gay et al., 2019).

-Fig. 3c – the quantification of organoid numbers will help the reader.

The quantification is featured in Fig. 3B (further quantifications for organoid experiments: Fig. S4C,D).

-Extended data Fig. 3b- could the authors provide 5 top markers of each cluster rather than showing known gene markers of each.

This information is now added in Fig.S5A.

Reviewer #3 (Remarks to the Author):

The authors employ conditional mouse models to dissect the different contributions of N-terminal and C-terminal -catenin binding partners to stem cell-driven homeostasis in the intestinal crypts. Selective loss of C-terminal interactions abolishes Wnt signaling, causing rapid loss of viable crypts. In contrast, selective loss of N-terminal interactions initially leads to hyperproliferation of Intestinal stem cells, followed by loss of stem cells through onset of differentiation programs likely driven by activation of JNK signaling. In general, the findings of this study are interesting, well-founded and broadens our mechanistic insights into Wnt-regulated stem cell maintenance and differentiation in the intestine.

Critique

-Maybe I missed this, but what are the possible (indirect) effects of the mutant alleles on cell adhesion? Could this be contributing to some of the phenotypes observed?

We had previously shown that β -catenin mutants are fully competent in adherens junctions (Valenta et al., 2011, Valenta et al., 2016). Now we provide additional evidence that the mutants maintain their structural role at the cell membrane. As shown in Fig.S1G mutant β -catenin proteins colocalize with Epcam at adherens junctions, and the adherens junctions appear unaffected. Moreover, the integrity and functionality of the intestinal barrier is not compromised up until 4d pi in mutant animals, as shown by FITC-dextran administration *in vivo* (Fig.5G). Since we show that the adherens junctions, the key constituents of the intestinal barrier, are morphologically intact and functionally uncompromised, we conclude that the observed phenotypes are the consequence of altered transcriptional outputs of β -catenin.

The newly added Fig.S1G

G) wt β -catenin, β -catenin- Δ C and β -catenin-D164A co-localize with epithelial cell adhesion molecule (Epcam) at the cell membrane. Timepoint: 2d pi. Scale bars, 20 μ M.

The newly added Fig.5G

G) Intestinal permeability of control, D164A and Δ C animals 2d and 4d pi, as quantified by FITC-dextran concentration (mg/mL) in plasma. Horizontal line indicates mean concentration. $p > 0.01$, as calculated by one-way ANOVA with Tukey's post hoc test ($n=2-3$).

-Ext fig1E – What are the total and nuclear β -catenin levels in the induced N/C mutants relative to controls?

We added the immunoblot panel (Fig.S1F) showing the nuclear and cytoplasmic levels of mutant and wt (i.e. flox allele) β -catenin protein. The levels are similar. The slightly reduced nuclear level of β -catenin- Δ C is most likely the result of the rapid loss of cells with active Wnt/ β -catenin signaling within the crypt (i.e. Wnt-receiving stem cells and early progenitors). These start to be lost at this timepoint and with them also the nuclear β -catenin. The comparable expression of β -catenin- Δ C, β -catenin-D164A and wt β -catenin (control) is also apparent from immunostaining at the adherens junctions (Fig. S1G). We are confident the β -catenin protein shown in these stainings is exclusively mutant β -catenin. Indeed, wt β -catenin is depleted in Δ C mutants 2d pi. For the immunostaining we used an antibody that recognizes the C-terminus and thus doesn't bind C-terminally truncated protein (Fig. S1E). Moreover, in the western blot, the lanes loaded with lysates obtained from crypt cells of Δ C mutant animals 2d pi only feature the shifted band of the Δ C mutant protein, and not of the wt protein (Fig. S1F).

-Is there any possibility that the mutant alleles can be indirectly influencing Tcf-catenin interactions due to altered folding?

We thank the reviewer for this comment. Altered folding influencing core armadillo repeats should be apparent by affected adherens junction due to the dual role of β -catenin. However, both β -catenin- Δ C and β -catenin-D164A are appropriately localized at adherens junctions (Fig.S1G), and the patterns of E-cadherin and Epcam are not affected (Fig.S1G, Fig1E and others). Since the binding sites for E-cadherin are similar/overlapping to those binding Tcf/Lef we do not expect impaired binding to Tcf/Lef or altered folding. Importantly the mutations introduced into used β -catenin mutants are not in the regions required for the binding to Tcf/Lef (Valenta *et al.*, 2011, Valenta *et al.*, 2016).

-Fig 1 – Use of *Lgr5* reporter mice to better characterize direct effects on the intestinal stem cells and their progeny would have benefitted the study. There is currently an over-reliance on *in situ* hybridization markers to generate conclusions regarding stem cell/progenitor populations.

We believe that using *Lgr5*-reporter strains may be problematic in our model system. Indeed, in the two most commonly used reporter strains *Lgr5-eGFP-IRES-CreERT2* (generated by the lab of H.Clevers) and *Lgr5-DTR-EGFP* (generated by the lab of F. J. de Sauvage), the transgenic alleles replace the endogenous *Lgr5* coding region. This results in transgenic animals that harbor one copy of functional *Lgr5* protein and are not homozygous viable. Because of reduced *Lgr5* levels, Wnt signaling may be altered (via de-regulated RNF43/ZNRF3). It would be challenging or impossible to distinguish this effect from the consequences of our mutants (i.e. mutant β -catenin and acute reduction of β -catenin levels). Hence, we believe that available *Lgr5*-reporter strains are not ideal to study perturbations and levels of Wnt signaling.

As an alternative approach we therefore used scRNASeq to “gate” for SCEP *in silico* and focus our analysis on stem cells and early progenitors, not only relying on *Lgr5* expression, but instead on a broader panel of stem cell and progenitor markers. We combined our single cell transcriptomic studies with extensive *in situ* hybridization and immunofluorescence, adding spatial and positional information to our analysis. For instance, we used smFISH-based labelling of *Lgr5*+ cells (Fig.1G), which was also used by many of the leading labs in the field, such as the labs of H.Clevers, A. van Oudenaarden and S. Itzkovitz. We believe that the combination of smFISH, and expression profiling (bulkRNAseq, single cell RNAseq and qRT-PCR) provide a solid spatial and quantitative data concerning the expression of *Lgr5*.

-Fig 1E – The levels of Axin2 and Olfm4 expression visualized on the control tissues appears markedly lower than expected. I would suggest this may be due to compromised tissue and needs repeating to obtain a fair comparison with the mutant crypts. As above, the use of Lgr5 reporter mice would have greatly simplified the stem cell analyses here.

We would like to point out that the appropriate control, throughout our study, is not the homozygous wt situation. It is instead a hemizygous situation that arises upon conditional deletion of one copy of β -catenin (*villin-CreER^{T2}; Ctnnb1-wt/flox*). This mutation is not lethal. Indeed, animals globally lacking one copy of β -catenin do not show any phenotype (Haegel et al., 1995). Using hemizygous animals as controls for Cre/loxP system is in general quite common. The effects of the acute reduction of β -catenin levels on Wnt signaling are thus expected, and apparent from our control time course scRNAseq data, where we saw a temporary and slight reduction in Wnt-target genes 2d pi, which is readily restored to unperturbed levels 4d pi (data not shown). However, in tissue sections of control animals 2d pi, *Axin2* mRNA and *Olfm4* are clearly expressed in the cells where they should be expressed. We did not use (and did not want to use) any level enhancement during post-processing (or high laser power during the collection of the data by confocal microscopy) which may also contribute to less intense signals.

-As I understand it, the study is restricted to evaluating the proximal SI only – are there region-specific differences observed?

Due to the high dynamics of the proximal small intestine we just focused on this part. However, the observed phenotype is not restricted to duodenum. Nevertheless, it is delayed along the proximal-distal axis. Whereas the jejunum phenotype is comparable to that of the duodenum, the colon does not seem to be affected as readily as the duodenum (added staining in Fig 3A-B) at the same time. As now mentioned in the text, this mirrors the delayed onset of Wnt/ β -catenin loss-of-function phenotypes in the colon that we previously observed (Valenta et al., 2016).

-What happens to Paneth cells @ early time-points following induction of the 164D allele? Given the changes in EphB expression observed, it would not be surprising if the Paneth cells quickly become mislocalised along the crypt-villus axis, potentially contributing to the observed stem cell phenotype.

Paneth cells do not seem to be altered in morphology, nor in proliferative signature, at earlier time point (2d pi) in the D164A mutant (please see new panels: Fig. S3C,D and Fig. S5F). As the reviewer suggested, at 4d pi, mislocalized and mis-differentiated cells can be found also in the villus and at the crypt/villus boundary (new panels: Fig.2B). Importantly, we also characterized the cells showing secretory mis-differentiation (mixed Paneth/goblet signature) by single cell RNAseq (Fig.S5E). However, only crypt cells were profiled by scRNASeq, so those intermediate cells reflect only the secretory hyperplasia in the crypts, where it is dominant, and not the mislocalization along the crypt-villus axis.

-Fig 1F – please confirm via q-PCR (or better via quantifying stem cell numbers using *Lgr5* reporter mice).

We performed experiments with more animals and added new real-time qRT-PCR panels (confirming and extending Fig.1F) – newly added Fig.S2C.

-Fig 1H – Plane of the control crypt is non-optimal – please repeat. Would it be worthwhile to more accurately quantify proliferation via EDU-injection/FACS (better yet, on sorted *Lgr5*+ stem cells and their progeny)?

As suggested, we repeated and substituted the control crypt shown in Fig 1H. Unfortunately, due to animal license restrictions, we were not allowed to perform EdU-injection experiments. Nevertheless, the proliferation activity in stem cells is extensively documented by various independent and orthogonal methods such as Ki67 staining (Fig.1E, and newly added staining in Fig S2F, Fig.2A), organoid-based experiments (Fig.3B-D), and most importantly cell cycle/proliferation analysis and enrichment of a Myc-E2F signature in the single cell RNAseq data of the stem cells and early progenitor cluster (Fig.4C,D and Fig.S5F).

-Fig 2 – Given the interplay between Wnt and Notch in determining stemness vs. differentiation and the fact that the enhanced secretory cell phenotype is reminiscent of blockade of Notch signaling using inhibitors, is there any indirect effect of the mutant alleles on Notch signaling (perhaps via the JNK signaling route)?

We thank the reviewer for this question. We did not investigate it in detail. However, we did not observe, and did not expect, any specific impact on Notch signaling in our β -catenin mutants. The secretory phenotype is a matter of an interplay between Wnt and Notch, and previous publications demonstrated that the secretory phenotype can be elicited not only by blocking Notch, but also by perturbing Wnt or Fgf pathways (reference 19-23 in the manuscript). Nonetheless, we checked and observed an upregulation of Notch1 in the N-terminal mutant animals at 2d dpi (bulk RNAseq data), suggesting that the Notch pathway is not repressed. On the other hand, *Notch1* mRNA levels are strongly reduced in D164A mutant crypts 4d pi, but this might reflect reduced number of SCEP.

-Fig 3 – The fact that the mutant organoids cannot be passaged implies that there is indeed a rapid loss of stem cells. What happens to the Paneth cells in these mutant organoids? Is the differentiation phenotype recapitulated in the organoids?

Unfortunately, we could not perform such type of experiments/analysis in organoids. The reason is that, in contrast to the *in vivo* situation, one copy of wt β -catenin (i.e. hemizygous state) is not sufficient to sustain the growth of the organoids. Even addition of external Wnt3a did not support the growth *villinCreER^{T2};Ctnnb1-wt/flox* organoids upon addition of tamoxifen. This is in striking contrast to the *in vivo* situation, where *Ctnnb1-KO/wt* animals are perfectly viable and fertile. Because of this discrepancy between the phenotype arising *in vivo* and the one arising *in vitro*, we focused our efforts to dissect the phenotypic changes occurring in the mouse. Other aspects of this issue are discussed in the replies to reviewer 1 and reviewer 2.

-Fig 3- Does Day 0 refer to organoids induced and then immediately harvested, or to non-induced organoids? If the latter, it is surprising that there is so little budding evident in the controls. If the former, I would then like to see non-induced organoids plated to generate an accurate comparison of the difference in organoid formation/budding extent. Again, the ability to selectively isolate *Lgr5+* stem cells from control/mutant reporter mice would have facilitated a more direct evaluation of functional effects on the stem cells via organoid

assay (and also a more accurate analysis of direct transcriptional changes within the stem cell compartment via ATAC-SEQ etc).

The day 0 in former Fig. 2B,C (now Fig.3B,C) refers to days post injection (so in this particular case non-injected crypts/organoids). Days in former Fig.2C (now Fig.3C) refer to days the crypts spent in culture after being isolated from animals 2d pi. As mentioned, the control is not homozygous wt, but *villinCreERT2*, *Ctnn1-wt/flox* (so hemizygous, after the tamoxifen treatment). Plating of homozygous wt organoids is in the newly added Fig.S4. We added a more accurate analysis of transcriptional changes and organoid quantification to document the death of organoids upon induction of recombination *in vitro* (Fig.S5).

The newly added Fig.S4B-E

B) *villin-CreERT2*; *Ctnnb1*^{wt/flox} and *villin-CreERT2*; *Ctnnb1*^{D164A/flox} organoids treated with 4-hydroxytamoxifen (4OHT) *in vitro* die 7 days after induction of recombination. Duodenal organoids lacking the *villin-CreERT2* allele are not affected by 4OHT. The number of days after addition of 4OHT are indicated at the top. C) Quantification of live organoids upon 4OHT addition. Abbreviations: het = *villin-CreERT2*; *Ctnnb1*^{wt/flox}, no Cre = *Ctnnb1*^{wt/flox}, D164A = *villin-CreERT2*; *Ctnnb1*^{D164A/flox}. Error bars indicate standard deviation (n=2). D) qRT-PCR of *Axin2* (Wnt target gene), *Ccnd1* (proliferative marker), and *Lgr5* (IESCs marker) genes in D164A and control organoids upon addition of 4OHT. Expression levels are normalized to GAPDH. Error bars indicate standard deviation (n=2). E) *villin-CreERT2*; *Ctnnb1*^{wt/flox} organoids grown in medium in Wnt3a-conditioned medium die 5 days after 4OHT addition. 20x magnification.

-Fig 3 – Given that the ability of isolated crypts to survive as organoids ex vivo is dictated by their ability to rapidly circularize after plating, is it possible that the mutants simply decrease the efficacy of this initial process due to reduced adhesion? If the organoids are first generated before inducing mutant expression, perhaps the phenotype would be less severe/different?

This information is now added to Fig.S4. As mentioned previously, the phenotype is not less severe, nor different. Death occurs even if recombination is induced in already established organoids. The additions of exogenous Wnt3a did not promote the survival (Fig.S4E).

-Fig 4D – why is Lgr5 not included here?

This is due to the sparsity of the data collected by the droplet based scRNAseq method used in this study (Dropseq). Low abundant transcripts are particularly affected by the low capture rate of this method. We provide other markers instead, and a more broad stem cell signature in Fig.4A,D. Additionally, smFISH of *Lgr5* is more reliable, quantitative and highly efficient for lowly expressed genes (Fig.1G). The real-time qRT PCR provides additional conformation/insight (Fig.S2C).

-Fig 5 – the link with JNK signaling is interesting- can this be validated by modulating this signaling pathway in intestinal organoids?

As discussed above in more detail, organoids do not represent a suitable system to address this. We believe that the evidence we show (*in vivo* data) provide good evidence.

Reviewer #4 (Remarks to the Author):

In this manuscript the authors investigated how Wnt/beta-cat signaling defines the transcriptional outputs onto stem cell/epithelial progenitors to regulate intestinal homeostasis based on co-factor usage. They utilized elegant mice models to discern the fine tuning of Wnt signaling mediated by C or N-terminally co activators of beta-cat that are able to govern Wnt output solely from mutated allele. Additionally, other reference mouse models were used such as dm mutant mice combining both N and C coactivator deletion and beta-cat KO. They conclude that C-terminally recruited activators act as on-off-switch for the beta-cat mediated transcription of Wnt target genes, while N-terminal cofactors fine tune wnt signaling in order to ensure proper proliferation and self-renewal of intestinal stem cells. Overall this work is timely and will be of interest to the Nat Com readership, however across the manuscript the data has been selectively presented, controls not properly annotated, statistics provisionally added. These shortcomings, in my opinion, raise major concern(s) and need to be critically addressed. Therefore my review, at this point, focuses mainly on concerns regarding consistency of data presentation (taking into an account the information authors provided).

Major concerns:

- In figure 1 authors conclude that expression profile of N- vs C-terminal beta-cat co-factors is distinct and expression profile of delta C is similar to KO based on analyses of bulk rna sequencing of multidimensional data set of several groups n=2 deltaC, n=3 D164, n=2 Ko, n=3 dm, n=3 control. This is a rather small n per group and yet authors choose to present only one sample of each group in Figure1B. The authors should better present all samples used for sequencing individually to reflect biological variance among the animals belonging to each group.

Concerning the bulkRNAseq data, the PCA graph shown in Fig.1B depicts the group average (not individual data points) as indicated in the legend. The Fig.1D (heatmap) provides an information concerning sample variance for marker genes which are relevant for assessing the activity of Wnt/ β -catenin signaling. Additionally, we added real-time qRT PCR validation with increased number of animals (Fig.S2C).

-On the following panel 1C the cross comparison of this multidimensional data set is presented. Can authors explain and justify (also in the manuscript) why dm mutant mice that are missing both N and C terminally co-activators have only around 50 DEGs. One would expect broader changes in these animals compared to delta C or D164A mutant.

We thank for this interesting question. We did not investigate this issue in detail, because we focused mainly on dissecting the differences between N. vs. C-terminal branches of co-activators. We used dm animals, whose phenotype we already described (Valenta *et al*, 2016), as a control for the situation in which the transcriptional outputs of β -catenin are completely abrogated. This is an even more appropriate control than the complete loss of β -catenin, as in the latter case, epithelial adhesiveness is additionally compromised. Also, as noted in the text, the number of DEG is also influenced by immune infiltrate in the D164A sample. However, the differences may also arise because the opposing effects of two branches of co-activators level each other out. Alternatively, in the individual Δ C-mutant there might be still a possibility for some compensatory mechanisms (via N-terminal branch) that, despite these mechanisms fail at the end, they might be reflected by the higher number of DEG's. Hence, our results point to the fact that blocking just the contribution of C-terminal co-activators is not the same as the complete block of β -catenin's transcriptional outputs.

-Additionally, the reference 17 does not refer to dm mutant as annotated in manuscript.

Actually, it does. In the Valenta *et al* 2016, it is mentioned "To specifically probe the contribution of the transcriptional output of canonical Wnt signaling, we used the β -catenin-dm allele (Valenta *et al.*, 2011)." In that publication we used the dm mutant (together with the complete l-o-f allele) and compared it to *Wntless* loss of function in the intestine. Data concerning the dm are depicted in the Supplementary Material of that reference.

-Please present the comparisons between groups as venn diagrams for easier visualization, this is very well achievable among 4 groups.

Venn diagram representation is now shown in Fig.S2A.

-The biological variance between the samples belonging to each group is obvious across the sample of genes presented in 1D heatmap. Clearly not all the genes change in the same direction within the same group (see mm7, sox9, tnfs19...). This leaves me wonder whether 2-3 animals per group are representative sample size that authors repeatedly use, even in downstream analyses such as in situ hybridisation. Secondly quantification of Rna dots by this particular technique is rather challenging if not arbitrary even with the software obtained from company. Thirdly, the statistics can not be performed on 2 biological replicates, therefore either include more animals or exclude the statistics.

Following this relevant point we excluded the statistics were just 2 biological replicates are used. On the other hand, we increased the number of biological replicates – for example as shown in Fig.S2C. We would like to stress the observed phenotypes were determined repeatedly, documented by independent, orthogonal methods, and we did not observe any animals without the phenotypic change. Importantly, for real time qRT-PCR and stainings/*in situ* hybridisations we used different animals than those for bulkRNAseq or scRNAseq data.

-As adding more biological replicates to sequencing data might be time consuming and expensive the authors should critically perform qRT pcr on at least 3-5 animals per group (including all the groups: dm, wt, deltaC, D164A, KO) for at least Lgr5, Olfm4, Ascl2, axin2 and cd44 and IHC for c-myc (and present those on the same graph).

As mentioned above, we performed additional experiments and increased the number of animals (Fig.S2C). We repeatedly tried to stain for cMyc by immunohistochemistry (using various antibodies). However, we failed. We discussed this issue with Prof. Bruno Amati (Department of Experimental Oncology, IFOM-IEO, Milano, Italy), an expert within the cMyc field. He confirmed, that staining for cMyc in primary intestinal tissue is very tricky and almost never works. We believe that the combination of smFISH with expression profiling (strong Myc-E2F signature enrichment) provide strong evidence concerning the changes in c-myc expression, both quantitatively and spatially.

-Delta C mutant in supplementary figure 1E appears to be very different than control at 2dpi. Multiple crypts are undergoing crypt fission, a sign of increased stem cell activity which contrast with conclusions of figure 1 showing loss of stem cells and proliferation at 2dpi. This inconsistency together with such limited number of animals used for assays aligns with previous concern questioning consistency of the delta C phenotype or reflecting inconstancy or recombination? Therefore, in my opinion, phenotypes of delta C and D164A mutants would need to further investigated to justify the author's conclusions.

We did not observe any fission across multiple stainings, as can be seen below (both histological stainings are ΔC 2d pi, two different animals), or in Fig.S1B and Fig.S3C.

No fission was observed in ΔC mutants 2 d pi

H&E staining of ΔC (*villin-CreER^{T2};Ctnnb1^{ΔC/lox}*) shows no fission at 2d (sections from two different animals are shown).

-In figure 3 the authors focus on D164A mutant and observe on day 4pi loss of stem cells accompanied with secretory hyperplasia judged by Lyz/Pas staining and conclude that aberrant stem cell activation at day 2 leads to aberrant secretory cell differentiation into intermediate double positive Lyz/Pas cells (Figure 2B). However in supplementary figure 4A this cell type is not annotated nor its presence acknowledge in manually selecting cell clusters. This seems odd as authors in material and methods state to use crypt fraction for single cell sequencing and from figure 2 observe the majority of the double positive Lys/pas cells in the crypt whose presence is even acknowledged in the main text of the manuscript. Please clarify to me how is it possible to observe an increase of manually adjusted Paneth and goblet cell clusters (composed of fully differentiated Paneth or Goblet cells respectively with no overlap in expression Supplementary Fig3B) within the crypts of D164A while the crypts are composed of double positive secretory progenitors (Fig 2B) that were not accounted for?

We thank the reviewer for this comment. Supplementary Figure S5 now includes the profiling of intermediate cells in our scRNASeq data. These cells indeed concomitantly express mixed features of Paneth and goblet cells (Fig.S5E). The absence of this information in the previous version of the manuscript was due to the fact that we focused our analysis on stem cells and early precursors, while the intermediate secretory cells status is apparent mainly in differentiated cells.

-From the text it is not obvious from the first read what animals are used as a control for the single cell sequencing experiment presented on figure 4 and supplementary figure 3. Please include the genotype and explain in main text/figure legend.

This information is now indicated in the figure legend.

-On figure 4A is obvious that only a selection of single cells is presented (SCEP account 25-50% of the all cell analyzed, Supplementary figure 3 C). Please clarify whether the z score in figure 4C was calculated from the whole data set or preselected cells and if later the case specify the criteria in detail.

It was calculated only from the SCEP subset. Comparative analysis revealed no change in the other cell types (Fig.S5F).

-Please provide better representation of *Creb3l3* rna in situ and add quantification of crypts expressing it. In supplementary figure only 1 crypt (from 2 shown) is expressing the transcript, while the other doesn't and the expressing crypt does not belong to the crypt villus unit therefore it is questionable at which plane of the crypt villus unit this crypt belongs to.

We changed the representation. All crypts are expressing *Creb3l3*. In addition, to support our data on increased UPR stress, we now provide real time qRT-PCR of *Hspa5* and *Ddit3* (ER stress markers) in Fig. 5B.

The newly added Fig.5AS2A.

A) smFISH shows villus-restricted localization of *Creb3l3* mRNA in control animals (*villin-CreERT²; Ctnnb1^{wt/flox}*) and its ectopic expression in D164A (*villin-CreERT²; Ctnnb1^{D164A/flox}*) crypts. Red and green dotted lines indicate crypt and villus area, respectively. Red and green insets show higher magnification of crypt and villus areas, respectively. Arrows indicate single mRNA molecules visible as dots. Timepoint: 4d pi. Scale bars: 20 μ M.

B) qRT-PCR indicates increased expression of UPR markers in D164A crypts with respect to control. Expression levels were normalized to GAPDH. Timepoint: 4d pi. **** $p < 0.0001$, ** $p < 0.01$, * $p < 0.05$ * as calculated by unpaired Student's T-test (n = 3-4).

-What control (genotype) was used for generating the data presented on figure 5?

This information is now added to the legend. It was *villinCreERT2, Ctnnb1-wt/flox*.

Reviewers' Comments:

Reviewer #1:

Remarks to the Author:

Comments on revised Nature Communications manuscript NCOMMS-20-19482A "Differential regulation of β -catenin mediated transcription governs identity of intestinal epithelial stem cells" by Costanza Borrelli et al.

In response to my previous comments the authors undertook a huge effort to add a substantial amount of new data and to re-write large parts of the manuscript. I am happy to acknowledge that the revised version of manuscript has greatly improved and that most of my concerns were more than satisfyingly addressed. There is one exception, though. This concerns the occurrence of intermediate cells. Despite replacing the original figure and labeling the Paneth and goblet cells I am not convinced that there is massive occurrence of intermediate cells as suggested by the authors and the scheme in Fig. 2C. The results of the scRNA-seq also suggest that at d4 pi the majority of cells is lineage marker negative and intermediate cell represent only a small fraction only (Fig. S5E). However this is a minor issue and whether or not the authors tone down their statements about intermediate cells will not affect the main conclusions of their work which are nicely supported by the remaining data.

Additional issues:

- line 237: incorrect reference to Fig. 5F (should be Fig. S5F?)
- line 263 and 278: incorrect references to Fig. S3E (does not exist) and S5E.
- the authors should mention Fig. S5G or remove it from the manuscript

Reviewer #2:

Remarks to the Author:

The authors provided a substantial amount of new data and should be commended for their effort during this difficult time. Most of the concerns raised in the first submission have been addressed sufficiently. I endorse the publication of this study.

Reviewer #3:

Remarks to the Author:

In general, the authors have done a good job in addressing major concerns. I am not convinced that their reasoning for not evaluating direct functional effects on stem cells using Lgr5 reporter mice due to altered Wnt signalling is valid, since this has never been shown and there are now Lgr5 reporter/Cre lines available that do not reduce endogenous Lgr5 expression levels. It is always dangerous to rely on marker expression alone to conclude specific effects on stem cells. However, I do not think this sufficiently dilutes the overall findings to preclude publication.

Reviewer #4:

Remarks to the Author:

The authors have successfully answered my concerns.

We would like to thank all reviewers for their positive view regarding the revised version of our manuscript. We are very pleased that their constructive criticism and recommendations allowed us to improve the manuscript the way it can be accepted for the publication. In the line with the reviewer's recommendations and editorial requests, we have now revised the manuscript the way it follows editorial instructions, formatting guidelines and addressing remain reviewer's concerns.

The text changes throughout the manuscript added to improve the clarity of the portrayed messages or following editorial guidelines are highlighted in blue.

We believe that all these changes can transform our manuscript to be ready for the publication.

Note : Reviewer's comments are in ***Bold Italics***

Reviewer #1 (Remarks to the Author):

In response to my previous comments the authors undertook a huge effort to add a substantial amount of new data and to re-write large parts of the manuscript. I am happy to acknowledge that the revised version of manuscript has greatly improved and that most of my concerns were more than satisfyingly addressed. There is one exception, though. This concerns the occurrence of intermediate cells. Despite replacing the original figure and labeling the Paneth and goblet cells I am not convinced that there is massive occurrence of intermediate cells as suggested by the authors and the scheme in Fig. 2C. The results of the scRNA-seq also suggest that at d4 pi the majority of cells is lineage marker negative and intermediate cell represent only a small fraction only (Fig. S5E). However this is a minor issue and whether or not the authors tone down their statements about intermediate cells will not affect the main conclusions of their work which are nicely supported by the remaining data.

Additional issues:

- line 237: incorrect reference to Fig. 5F (should be Fig. S5F?)
- line 263 and 278: incorrect references to Fig. S3E (does not exist) and S5E.
- the authors should mention Fig. S5G or remove it from the manuscript

We thank the reviewer for acknowledging the revised version of our manuscript. We corrected all additional issues (concerning figures). Additionally, we tuned down the statement concerning the intermediate cells. Newly added statement (highlighted in blue) to the relevant part of Result section: *Moreover, we observed Lyz⁺ cells mislocalized to the villus of D164A animals (Fig. 2b). These cells are reminiscent of intermediate cells, which share both Paneth (lysozyme) and goblet cell traits (mucins), are the result of stem cell mis-differentiation, and have been observed upon experimental perturbation of Notch, Wnt or EGF signalling in the intestinal epithelium²²⁻²⁶*

Reviewer #2 (Remarks to the Author):

The authors provided a substantial amount of new data and should be commended for their effort during this difficult time. Most of the concerns raised in the first submission have been addressed sufficiently. I endorse the publication of this study.

We would like to thank the reviewer for endorsing our revised manuscript for the publication.

Reviewer #3 (Remarks to the Author):

In general, the authors have done a good job in addressing major concerns. I am not convinced that their reasoning for not evaluating direct functional effects on stem cells using Lgr5 reporter mice due to altered Wnt signalling is valid, since this has never been shown and there are now Lgr5 reporter/Cre lines available that do not reduce endogenous Lgr5 expression levels. It is always dangerous to rely on marker expression alone to conclude specific effects on stem cells. However, I do not think this sufficiently dilutes the overall findings to preclude publication.

We would like to thank the reviewer for positively evaluating our revised manuscript. We agree that using Lgr5 reporters that do not alter Lgr5 function would be beneficial. Based on this fact and reviewer's recommendation we now acknowledged the limits of our marker expression analysis connected to not using Lgr5-reporter. Newly added statement (highlighted in blue) to the relevant part of Result section: *As our mouse models don't harbor any stem cell reporter (eg. Lgr5-EGFP) that enables identification of stem cells, we tested this hypothesis computationally. We trained a logistic regression model on TAs and IESCs obtained from a publicly available single cell dataset of murine small intestinal epithelium³², and tested SCEP cells from 0d and 2d pi against this model.*

Reviewer #4 (Remarks to the Author):

The authors have successfully answered my concerns.

We thank the reviewer for the positive view concerning our revised manuscript.